



# Plio-Quaternary tectonic evolution of the southern margin of the Alboran Basin (Western Mediterranean)

Manfred Lafosse[1,*], Elia d'Acremont[1], Alain Rabaute[1], Ferran Estrada[2], Martin Jollivet-Castelot[3], Juan
Tomas Vazquez [4], Jesus Galindo-Zaldivar [5,6], Gemma Ercilla[2], Belen Alonso[2], Abdellah Ammar[7],
Christian Gorini [1]

[1] Sorbonne Université, CNRS-INSU, Institut des Sciences de la Terre Paris, ISTeP UMR 7193, F-75005 Paris, France
[2] Instituto de Ciencias del Mar, ICM-CSIC, Continental Margin Group, 08003 Barcelona, Spain
[3] Univ. Lille, CNRS, Univ. Littoral Côte d'Opale, UMR 8187, Labratoire d'Océanologie et de Géosciences (LOG), F59000, Lille, France
[4] Instituto Espanol de Oceanografia, C.O.Malaga, Fuengirola, Spain
[5] Dpto. de Geodinamica, Universidad de Granada, Granada, Spain.
[6] Instituto Andaluz de Ciencias de la Tierra (CSIC-UGR), Granada, Spain.
[7] Université Mohammed V-Agdal, Rabat, Morocco
[*]now at: Tectonic and Structural Geology Groups, Department of Earth Sciences, Utrecht University, PO Box 80.021, 3508 TA Utrecht, The Netherlands

Correspondence to: Manfred Lafosse (m.r.lafosse@uu.nl)

**Abstract.** Progresses in understanding the sedimentary dynamic of the Western Alboran Basin lead us to propose a model of evolution of its tectonic inversion since the Pliocene to present-time. Extensive and strike-slip structures accommodate the Miocene back-arc extension of the Alboran Basin, but undergo progressive tectonic inversion since the Tortonian. Across the Alboran Basin, the Alboran Ridge becomes a transpressive structure accommodating the shortening. We map its southwestern termination: a Pliocene rhombic structure exhibiting series of folds and thrusts. A younger structure, the Al-Idrissi fault zone (AIF), is Pleistocene to present-day active strike-slip fault zone. This fault zone crosses the Alboran Ridge and connects southward to the transtensive Nekor Basin and the Nekor fault. In the Moroccan shelf and at the edge of a submerged volcano, we date the inception of the local shelf subsidence from the 1.81-1.12 Ma. It marks the propagation of the AIF toward the Nekor Basin. Pliocene thrusts and folds and Quaternary transtension appear at first sight as different tectonic periods but reflects the long-term evolution of a transpressive system. Despite a constant direction of Africa/Eurasia convergence since 5Ma at the scale of the southern margin of Alboran Basin, the Pliocene-Quaternary inversion evolves from transpressive to transtensive on the AIF and the Nekor Basin. This system reflects the expected evolution of the deformation of the Alboran Basin under the indentation of the African lithosphere.



## 1. Introduction

In brittle regime, oblique compression leads to strain partitioning between lateral motion and efficient rock uplifts (Fossen et al., 1994; Fossen and Tikoff, 1998). With time, the simple shear deformation involves blocks rotation and changes in the local stress field leading to the formation of better oriented tectonic structures (Nur et al., 1986; Ron et al., 2001; Scholz et al., 2010). It often results in a complex pattern of distributed deformation with transpressive and/or transtensive structures. The Alboran Basin is a typical example of such a complex tectonic evolution.

The Alboran Basin develops over a collapsed Tertiary orogen and corresponds to internal units of the Betic-Rif belt (Fig. 1) (Comas et al., 1999), and together form what we call the Alboran tectonic domain. The formation of the Alboran Basin has been linked to back-arc extension and lithospheric tearing during early Miocene (e.g. Jolivet et al., 2009, 2008). The Africa-Eurasia NW-SE oblique converging regime takes control during the Late Miocene, leading to a tectonic reorganization into a transpressive deformation pattern (Comas et al., 1999; Do Couto et al., 2016). DeMets et al. (2015) showed recently that it is possible to constraint very precisely the location of the rotation poles between Eurasia, North America, and Africa since the Miocene. The migration of the rotation pole between Africa and Eurasia toward the SE during the Pliocene and the Quaternary results in a roughly constant direction of Africa-Eurasia convergence, an increase in the convergence rates from approximately ~3.5 mm/y to ~5.5 mm/y at 35° N / 5° W between 5.2 Ma and present-day, respectively (DeMets et al., 2015). More recently, Spakman et al., (2018) show that the result of Africa –Eurasia absolute convergence is the retreat of the Gibraltar slab and the relative motion of Africa and Eurasia of 15km in the NNE-SSW direction for the last 8 Ma. Several strike-slip shear zones running from the Iberian to the Moroccan margins accommodate this convergence since the Miocene forming a broad shear zone called the Trans-Alboran Shear Zone (TASZ; Fig. 1)(Leblanc and Olivier, 1984). Following the westward slab retreat during Miocene time, the TASZ behaves as a left-lateral transfer fault zone accommodating the extension of the Alboran Basin. Due to ongoing Africa-Eurasia convergence, the TASZ underwent an oblique positive inversion starting around the 8-7 Ma Tortonian stage in the Betic Margin of the Bas Basin (Do Couto et al., 2014; Martínez-García et al., 2017). The compression migrates westward in relation with the slab roll-back, since approximately 7 Ma from the Algerian margin to the Alboran Ridge, and since ca. 5 Ma on the Al-Idrissi fault (Fig. 1 and 2) (Giaconia et al., 2015). From the Late-Miocene, several methods show the simple shear deformation of the Alboran Basin. Using a block rotation pinned model, Meghraoui and Pondrelli, (2013) have demonstrated for example that oblique convergence can lead to a rigid block rotation which can be accommodated by transcurrent faults (e.g. the TASZ, Fig. 1). Changes in stress direction are demonstrated in the Betic-Rif belt during the Plio-Quaternary (Aït Brahim and Chotin, 1990; Galindo-Zaldívar et al., 1993; Giaconia et al., 2015; Martínez-Díaz and Hernández-Enrile, 2004). The change in horizontal stress directions has led to compression and uplift of Plio-Quaternary sediments offshore the Palomares fault on the Iberian Margin (Giaconia et al., 2015). At present-time, the direction of shortening seems orthogonal to the NE-SW structures of the TASZ (Fig. 1)(Palano et al., 2013)

The distribution of the seismicity in the western part of the Betic-Rif belt reveals complex geodynamic interactions. Deep earthquakes occur at depths >60 km (Fig. 2a). They are located in the central Betic, beneath the West Alboran Basin, and the



Rif Mountains (WAB, Fig. 1), and are associated to a sinking slab (Fig. 2a) (Bezada et al., 2013; Ruiz-Constán et al., 2011; Thurner et al., 2014). In addition to the Africa-Eurasia convergence, this lithospheric scale process such as mantel delamination can have a strong influence on the deformation and the structure of the Alboran Basin (Petit et al., 2015; Thurner et al., 2014). The mechanical coupling between the Alboran Domain and the subsiding lithosphere (e.g. Perouse et al., 2010; Neres et al., 2016) , and/or  slab dragging under Africa/Eurasia convergence (Spakman et al., 2018) could cause the extrusion the Betico-

Rifian belt toward the South-West (e.g. Petit et al., 2015; Thurner et al., 2014). Because of those concomitant processes, the Africa-Eurasia plate boundary in the Alboran Basin and in the Betic - Rif belt cannot be assigned to a single fault system (Fadil et al., 2006) and some authors proposed to define a diffuse plate boundary between Africa and Eurasia (e.g. Palano et al., 2015). At the crustal level, recent progress in structural mapping have shown that the distribution of the deformation in the Alboran Sea switched from the Tortonian NE-SW to Quaternary NNE-SSW faults (Estrada et al., 2018; Galindo-Zaldivar et

al., 2018; Lafosse et al., 2017; Martínez-García et al., 2013, 2017). However, the timing and the mechanism of this structural evolution remains poorly constraint.

In the present work, we address the issues of structural evolution through the Plio-Quaternary of southwestern margin of the Alboran Sea toward the termination of the TASZ. We analysis of multi-resolution seismic reflection data, TOPAS profiles, and multibeam data. This structural subdivision may reflect a Quaternary change in tectonic style which reveals the present-

day strain partitioning of the deformation (Lafosse et al., 2017; Neres et al., 2016). Based on these recent datasets, we propose a new tectonic model explaining how the Africa-Eurasia oblique convergence can control the geometry of the tectonic structures of the southern margin of the Alboran Basin.

### 1.1.    Geological and geodynamical settings

In the southern margin of the Alboran Sea, the main structural element corresponds to the Alboran Ridge. The Alboran Ridge

divides the Alboran Basin in three different sub-basins: the Western Alboran Basin (WAB), the South Alboran Basin (SAB), and the East Alboran Basin (EAB) (Fig. 1). Transpressive and transtensive structures associated to the Alboran Ridge and to the Yusuf fault zone, respectively, as well as to several volcanic and/or metamorphic highs limit those sub-basins (Fig. 1). The Alboran Ridge is divided by the AIF (Fig. 1) into the South Alboran Ridge (SAR, Fig. 1), which corresponds to the submarine highs striking in the NE-SW direction (Xauen Bank, Petit Tofino Bank, Tofino bank, Ramon Margalef High, Eurofleet High,

Francesc Pages High, Fig. 3) and the North Alboran Ridge (NAR; Fig 1). The Alboran Ridge could correspond to a tectonic high building up since the Late-Miocene and experiencing several phase of deformation (Bourgois et al., 1992; Do Couto, 2014). The most recent one involves sinistral motions along a recent NNW-SSE transtensive fault network, the sinistral Al-Idrissi strike-slip fault, and the front Al-Indentation of the northern part of the Alboran Ridge (Estrada et al., 2018). The AIF connects to the south to the transtensive Nekor Basin (Lafosse et al., 2017).

Volcanism and tectonic deformation shaped the morphology of the southern Alboran Basin. The SAR is 70 km long and corresponds to a series of faults and folds affecting the Plio-Quaternary depositional sequences (Fig. 3)(Bourgois et al., 1992; Chalouan et al., 1997; Gensous et al., 1986; Martínez-García et al., 2013; Muñoz et al., 2008; Tesson et al., 1987) and to a



succession of submarine highs culminating around -110m (Fig. 3). The southern front of the SAR corresponds to the northern flank of a large NE-SW syncline called the South Alboran Trough (Fig. 3). The northern front of the SAR corresponds to the

Alboran Channel and the WAB (Fig. 3). In relation to back-arc extension or mantellic delamination, the SAR marks the transition from a thinned continental crust to the north to thick continental crust to the southwest (Díaz et al., 2016). Local occurrences of volcanism in the Francesc Pagès Bank and in the Ras Tarf are of Miocene age. The volcanism in the Francesc Pagès Bank is not accurately dated (Gill et al., 2004) but the lithology corresponds to basaltic rocks dated between 9.6 and 8.7 Ma in the same area by (Duggen et al., 2004). In the Ras Tarf, the volcanism ends around 9Ma (El Azzouzi et al., 2014).

According to Giaconia et al., (2015), since the Late-Miocene, the deformation has migrated toward the South West; the SAR might have accommodated a more recent left lateral shear than the NAR and the Carboneras Fault during the Messinian/early-Pliocene times (Fig. 2). Do Couto et al., (2016) proposed that the SAR is compressive since 8 Ma in association with the left-lateral strike-slip of the Carboneras fault zone (Fig. 1 and 2). The SAR is Miocene extensive structure, but E-W folds over north and south dipping thrusts accommodates the shortening of the Alboran Basin and demonstrate a tectonic inversion (Fig.

2)(Chalouan et al., 1997). As evidenced by the seismic reflection data, under-compacted shales deposited during the early to mid-Miocene extensive period are present at the bottom of the sedimentary column west of the SAR (Do Couto, 2014; Do Couto et al., 2016; Soto et al., 2008).

On both flanks of the SAR, the contourite deposits produce significant thickness variations of the Quaternary depositional units, that are pinched and thinned toward the foot of the submarines highs (Juan et al., 2016). Pre-Messinian deposits are

exposed at the seafloor in the core of the anticlines (Chalouan et al., 2008; Do Couto et al., 2016; Juan et al., 2016; Tesson et al., 1987). Seismic reflection profiles and well data show that the compressive folding continued until the Quaternary in the Francesc Pagès Bank and highlight several erosion periods during Plio-Quaternary time (Galindo-Zaldivar et al., 2018; Tesson et al., 1987).

The present-day deformation of the southern margin of Alboran Basin is accommodated along the AIF, a shear zone crossing

the NAR and the SAR at the NE tip of the Francèsc Pagès Bank (Fig. 2 and 3) (Dillon et al., 1980). Bathymetric and seismic reflection data have shown that the deformation along the AIF is accommodated through a series of sinistral NNE-SSW strike-slip faults segments (Fig. 1 and 2)(Ballesteros et al., 2008; Martínez-García et al., 2011). The AIF propagated southward during the Quaternary (Ballesteros et al., 2008; Gràcia et al., 2006; Martínez-García et al., 2011, 2013), connecting to the NNE-SSW active strike-slip faults north of the Al Hoceima region at the Boussekkour - Bokoya fault zone (d'Acremont et al., 2014;

Calvert et al., 1997; Lafosse et al., 2017). Above the Messinian Erosional Surface (MES) (Estrada et al., 2011; Garcia-Castellanos et al., 2011), the deep sedimentation in the Alboran Sea is driven by contouritic processes that also shape the seafloor since 5.33 Ma (Ercilla et al., 2016; Juan et al., 2016). Marine erosion can occur at the moat of the contouritic systems, generally at the foot of the slopes whereas deposition occurs at deepest locations Ma (Ercilla et al., 2016; Juan et al., 2016).

The rotation of the far-field direction of convergence from the Messinian Salinity Crisis (MSC) period to the present-day can

trigger three phases of deformation along the Alboran Ridge (Martínez-García et al., 2013). Unconformities and increasing accumulation rates demonstrate three tectonic phases: a tectonic phase-1 dated from 5.33 Ma to 4.57 Ma, a tectonic phase-2



from 3.28 Ma to 2.45 Ma and a last tectonic phase-3 between 1.81 Ma and 1.19 Ma. GPS kinematics shows a WNW-ESE relative displacement at a slip rate of 4.2 mm/y (Nocquet and Calais, 2004). More recently it has been suggested that the uplift along the Alboran Ridge culminated around 2.45 Ma in response to tectonic inversion (Martínez-García et al., 2017).

At present day, GPS velocities define an Alboran tectonic domain in between African or Iberian rigid blocks (Neres et al., 2016; Palano et al., 2013, 2015). This block is limited eastward by the TASZ and by the Yusuf Fault (Fig. 1 and 2b). East of the TASZ, the region corresponds to the SAB and the Oriental External Rif which behave as the African block (Koulali et al., 2011; Vernant et al., 2010). From GPS data, the maximum present-day rates of extrusion of the Alboran tectonic domain are close to 5.5-6mm/y between the Jebha and Nekor faults (Koulali et al., 2011; Vernant et al., 2010). These geodetic data show

a maximum southwestward lateral escape localized between the Nekor fault and the SAR-Jebha Fault area (Fig. 2).

In the SAB, the AIF and the Nekor Basin are affected by important crustal seismicity (Bezzeghoud and Buforn, 1999; Stich et al., 2005). In the area of the AIF, the earthquakes mainly occur at depth above 30km (Buforn et al., 2017). In the Nekor Basin, the seismogenic depth interval is between 0 and 11km depth (Van der Woerd et al., 2014). The 1994 and 2004 earthquakes in the Al-Hoceima area reached Mw=6.3 and 5.9 respectively (Fig. 4) (Custódio et al., 2016). On January 25th, 2016, an

earthquake further localized in the vicinity of the AIF zone reached Mw=6.3 (Buforn et al., 2017; Medina & Cherkaoui, 2017; Galindo-Zaldívar et al., 2018). The focal mechanisms of those three main regional earthquakes show sub-vertical nodal planes and a left lateral displacement (Fig. 4)(Bezzeghoud and Buforn, 1999; Biggs et al., 2006; Calvert et al., 1997; El Alami et al., 1998; Hatzfeld et al., 1993; Stich et al., 2005, 2006). Near the offshore Nekor Basin, close to the Moroccan coast, the NNE-SSW fault tracks identified at the seafloor in the vicinity of the epicenters are in accordance with the active fault planes deduced

from seismological data (d'Acremont et al., 2014; Calvert et al., 1997; Lafosse et al., 2017). In the deep basin, the January 25th 2016 earthquake sequence indicates a strike-slip style parallel to the trend of the AIF, with mainly NNE-SSW left-lateral motion (Ballesteros et al., 2008; Buforn et al., 2017; Galindo-Zaldivar et al., 2018; Martínez-García et al., 2011; Medina and Cherkaoui, 2017). The Alboran Ridge is reactivated near the AIF, as shown by several compressive focal mechanisms with NE-SW nodal planes parallel to the Alboran Ridge thrust axis, and by strike-slip focal mechanisms with a left-lateral motion

(Fig. 4). In the Nekor Basin, the deformation is distributed into a normal component into the center of the basin and a left-lateral component in its boundaries (Fig. 4)(Lafosse et al., 2017). In the SAR, the style of the deformation is unclear with focal mechanisms showing indiscriminately strike-slip or normal components (Stich et al., 2010).

## 2. Material and methods

### 2.1. Data

The data used in this study consists of multichannel profiles, SPARKER and TOPAS profiles and multibeam bathymetry. They were acquired during three oceanographic surveys (Fig. 3). The seismic reflection data were acquired with a 12-channel-streamer during the Marlboro-1 survey in 2011, as eight NNW-SSE parallel lines crossing the W-E folds of the SAR and two WSW-ENE parallel lines in the southern domain (Fig. 1, 2 and 3). During the SARAS survey in 2012 (d'Acremont et al.,





2014; Lafosse et al., 2017; Rodriguez et al., 2017), were obtained SPARKER and TOPAS profiles, multibeam bathymetry and acoustic reflectivity at a 25m/pixel resolution of the deep submarine seafloor were acquired. During the MARLBORO-2 survey in 2012 (d'Acremont et al., 2014; Lafosse et al., 2017) SPARKER profiles and shallow multibeam bathymetry at a 5m/pixel resolution were acquired. The bathymetric data from the INCRISIS survey were also used (Galindo-Zaldivar et al., 2018). In addition, we use a Digital Elevation Model downloaded from the EMODNET data set (http://www.emodnet.eu/) to fill the missing parts of our dataset.

## 2.2.  Methods

We used the seismic reflection and TOPAS data interpretation to do the tectonic mapping of the subsurface. At the seafloor, we made a visual recognition of fault scarp using the multibeam bathymetry and the curvature maps. The curvature is known as a relevant parameter to track the fault offsets on 3D seismic section (e.g. Roberts, 2001) and at the seafloor (e.g. Paulatto et al., 2014). The seismic-stratigraphic analysis of the Plio-Quaternary sequences is based on the stratigraphy defined by Juan et
al., (2016). The sum of the plan curvature values was made with the help of ArcGis V10.2 using the focal statistics tool in order to remove the smooth the noise at depths greater than -150m. The chronology of the seismic stratigraphic boundaries was defined based on an age calibration on data from scientific wells DSDP 121 and ODP 976, 977, 978 and 979 (Fig. 1 and 5) (Ercilla et al., 2016; Juan et al., 2016). We consider an average P-wave velocity of 1800m/s for the Plio-Quaternary pelagic sedimentation (Soto et al., 2012). We propose seismic stratigraphy and sequential stratigraphy interpretations of depositional
units based on the nomenclature and general principles presented in the literature (Catuneanu, 2007; Catuneanu et al., 2011).

## 3.  Results

### 3.1.  Plio-Quaternary seismic stratigraphy

The Plio-Quaternary sedimentary register of the southern margin of the Alboran Sea has been divided into three Pliocene (Pl1, Pl2 and Pl3) and four Quaternary (Qt1 to Qt4) seismic units, based on the Juan et al. (2016) seismic stratigraphy (Fig. 5). These
units are limited at the bottom by discontinuity surfaces, M, P0 and P1 for the Pliocene units, and BQD, Q0 to Q2, for the Quaternary units. Boundaries represent discontinuity surfaces mostly defined by onlap and erosive surfaces; locally downlap surfaces are identified (Fig. 6 and 7). Sub-parallel, parallel, oblique and wavy stratified reflections characterize the Plio-Quaternary units.  Pl1, Pl2 and Pl3 units are pinching toward the structural highs and show aggrading wedges geometries. The Quaternary seismic units (QT1 to QT4) show an aggradational geometry and are confined to the foot of the folds where they
pinch on the older tilted Pliocene deposits (Fig. 6 and 7)(Juan et al., 2016). Contouritic deposits and associated sedimentary features, MTDs and volcanic deposits constitute the Plio-Quaternary units. The plastered drifts type is dominant and contribute to cover the structural highs (Juan et al., 2016). In seismic reflection, evidences of truncation at the foot of topographic high correspond to contourite moats and channels (Fig. 7). Sediments present local intercalations of lenticular chaotic or transparent facies interpreted as mass-flow deposits and correspond to scars on the bathymetry (Fig. 3)(Rodriguez et al., 2017). Regarding





the volcanic deposits, two buried volcanic edifices are identified on seismic reflection: the Big Al-Idrissi Volcano (Fig. 3, 8 and 11), and the Small Al-Idrissi Volcano (Fig. 6, 9 and 11). Acoustically, they correspond to a seismic facies of poorly continuous high amplitude reflectors (Fig. 9). Pliocene to Quaternary reflectors onlaps on these seismic units (Fig. 8 and 9). The Big Al-Idrissi Volcano corresponds to a conic structure located to the North of the Ras Tarf, with a 4-8km wide( Fig. 8) that has been interpreted as a N-S volcanic ridge in Bourgois et al. (1992). The top of this seismic body merge with the M-

Reflector (Fig. 8). Plio-Quaternary seismic units showing prograding to aggrading sigmoid reflectors characterizes the growth of a continental shelf above the M reflector burying this volcano (Fig. 8). On the west side of this seamount, the trajectory of the offlap breaks is concave up indicating that the rate of progradation decreases progressively with time. Reflectors onlaps terminations on the bottomset and foresets of the prograding seismic units mark the beginning of a retrogradation after 1.81Ma (Fig. 8). West dipping normal faults offset the depositional unit of prograding sigmoid reflectors (Fig. 8). Those normal faults

correspond to scarps at the seafloor (Fig. 8 and 11). Toward the top of the sequence, the bottomset of the late-Pleistocene Moroccan shelf offshore of the Ras Tarf corresponds to a unit of flat lying reflectors (Fig. 8). The flat top of the Big Al-Idrissi volcano culminates at an approximate depth of 150-200 m below the present-day sea level and corresponds to a toplap surface (Fig. 8).

The Small Al-Idrissi volcano at the South Alboran Through shows a roughly NNE-SSW spatial extent and a 4-5 km wide

conic structure (Fig. 9, 10 and 11). This seismic body intercalated within the Pl1 seismic unit (Fig. 9) corresponds to rounded high at the seafloor narrow submarine high (Fig. 11) which pinches abruptly toward the West (Fig. 9). The top of the Pl1seismic unit rests uncomfortably on this seismic body indicating an early-Pliocene age (Fig. 9). In the Fransesc Pagès Bank, a seismic body with similar facies is present at the core of an NNE-SSW striking anticline (Fig. 6), truncated by the M reflector (Fig. 6). The stratigraphic architecture of the shelf north of the Nekor Basin records an early-Quaternary regression (Fig. 12). We follow

the Q1 surface northeastward toward the top of the submerged shelf surrounding the Big Al-Idrissi Volcano (Fig. 8). The Q0 reflector corresponds to an unconformity at the bottom of prograding oblique reflectors. This depositional unit displays geometry of continental shelf deposits. The most distal offlap break shows the maximum extent of the continental shelf north of the Nekor Basin during the Pleistocene. It indicates that the retrogradation of the shoreline starts before 1.12 Ma and after 1.81Ma (Q1 reflector, Fig. 12). The most distal offlap break near Al-Hoceima is located around 312±30 mstwt corresponding

to a depth of 188±5 m below sea level (Fig. 12). In the distal part of the shelf, we interpret a seismic body of poorly continuous wavy reflectors deposited above an erosional surface as a local mass transport complex, which could mark an early Quaternary destabilization of the shelf.

### 3.2. Evidence and style of the compressive deformation

The seismic stratigraphy shows that the Plio Quaternary sequence records two major phase of deformation. Folds and faults

along the Alboran Ridge record the compressive phase and corresponds to a regressive trend on the Moroccan shelf. The second phase is younger and corresponds to local transgressions along the Moroccan shelf in relation with the activity of strike-slip and normal faults.





### 3.2.1.    Folded structures of the South Alboran Ridge (SAR)

The SAR region from the WAB to the South Alboran Trough corresponds to a wide 80-km long NE-SW folded area (Fig. 6).
The shortening in the SAR is distributed from east to west over the 10 to 25 km wide SAR structure, composed of a series of
two to four 4 km-wavelength anticlines (Figs. 6 and 10). Northward-verging anticlines characterize the northern front of
deformation (Fig. 6). In the eastern part of the SAR, the Francesc Pagès and the Eurofleet Highs correspond to pinched
anticlines, in a 10 km narrow fold, over south-verging thrusts (MAB16 and 14; Fig. 6). The anticline geometries underlie
several southward and northward dipping blind thrusts affecting the geometry of the M reflector (Fig. 6). From East to West,
series of anticlines and synclines draw a sigmoidal pattern (Fig. 10). Above the thrusts, a sigmoidal shape and left–lateral
deflections of their hinge axis (Fig. 6 and 10) characterize the Pliocene fold axes. The western part of the SAR corresponding
to the Xauen Bank is deflected toward the North (Fig. 6).

Below the M surface, truncated Miocene seismic units show local folding (Fig. 6f)(Do Couto et al., 2016). It indicates that a
shortening occurred in the SAR prior to the MSC. The lateral and vertical strata pattern of the Plio-Quaternary units show that
the shortening varies with time. Deformation are more important in Pliocene strata than in the Quaternary ones (Fig. 6). It
mainly striking when sediments are close to the tectonic features (folds and faults) and decreases laterally. Along the northern
flank of the SAR, tectonic tilting and to the growth of the drift deposits in relation with contourite system result in P0 to BQD
unconformities (Fig. 7). The intra-Pliocene unconformities, the tilting of the Pliocene units and the aggradation of Quaternary
contouritic deposits indicate a compressive deformation ending around the early Quaternary (Fig. 6 and 9). Within the Pliocene
sequence, the folding appears to be progressive and diachronic from East to West. At the foot of the Francesc Pagès Bank, P1
reflectors are unconformably lying on the P0 reflector (Fig. 7a). At the foot of the Ramon Margalef High, Pliocene reflectors
older than P1 show a more even geometry with constant thickness, P0 is likely to conformable (Fig. 7b).

Parallel to the SAR, the South Alboran Trough corresponds to large syncline that narrows from East to West (Fig. 6). Its
northern flank is steeper than the southern one (Fig. 6). The local variations of thicknesses reveal non-cylindrical folding of
the syncline (Fig. 6 and 10). The progressive tilt of the QT1 to QT4 units and internal growth strata reveal a more continuous
Quaternary to Pleistocene folding of the South Alboran Trough (Fig. 6, and 9) near the Al-Idrissi fault zone (Fig. 9). It indicates
that local folding persists during the Quaternary.

### 3.2.2.    The Al-Idrissi fault zone

At present day, the AIF is an NNE-SSW fault zone with several segments following older structural trend (Fig. 11). Crossing
the eastern end of the Francesc Pagès Bank and the western end of the NAR, it forms a positive flower structure distinct from
the Pliocene thrust of the Alboran Ridge (Fig. 13). The flower structure corresponds to a compressive bend of the AIF affects
the most-recent Quaternary sediments (Fig. 11 and 13), whereas Pliocene thrusts appears to be abandoned during the
Quaternary (Fig. 13). The depth of the Messinian unconformity at the western tip of the Alboran Ridge is lower than at the



Francesc Pages Bank (Fig. 10), indicating different uplift/subsidence rates from either part of the AIF. The location of the

compressive bend is highlighted at present-day by the cluster of compressive focal mechanisms (Fig. 4) (Stich et al., 2010). The southern tip of the AIF is characterized by NNE-SSW fault segments interpreted as splay faults distributing the deformation through the present-day deposits (Fig. 9 and 11). Below the volcanic facies, the poor acoustic penetration prohibits the interpretation of tectonic structures (Fig. 9). At the seafloor, the fault tracks are clear toward the southwest where they offset the Small Al-Idrissi volcano and link to the Bokoya fault (Fig. 10 and 11).

Affecting the SAR, N145° striking lineaments at the seafloor correspond to sub-vertical normal faults affecting the sub-surface sediments (Fig. 11 and 14). At the northern flank of the SAR, the fault network describes a 10-12km wide shear zone (Fig. 14). The recognition of pockmarks at the seafloor and signal attenuation near the faults on the seismic reflection data, which suggest fluid seepage in relation with active faulting (Fig. 14)(Judd and Hovland, 2009). Northward, the faults disappear at the seafloor under the present-day depositional part of the contourite. In the subsurface, they affects Q1 and Q2 surfaces

demonstrating late Pleistocene activity (Fig. 14b). Southward, the fault tracks are lost against the hinge axis of the Francesc Pagès fold. At the southwestern flank of the Francesc Pagès fold, similar NW-SE striking faults affect the seafloor (Fig. 11a). These N145° lineaments observed at the surface correspond to the normal faults pointed in red on the TOPAS profile (Fig. 6b), that uplift the western block of the fault wall. Despite poor expression at the seafloor, this fault zone continues southeast where it affects the whole Plio-Quaternary sequence (Fig. 9). Along the AIF, vertical offset of the P0 surface is around 101m

(Fig. 9). Between the N145° faults and the AIF, several fault segments affect the subsurface highlighting the distributed deformation that occurs between the N145° faults and the AIF with a more important vertical offset along the AIF (Fig. 9).

## 4.   Discussion

Our results show that the tectonic evolution of the Southern margin of the Alboran Sea evidences at least two phase of tectonic activity from the Early-Pliocene to present day. The first phase starts probably during the Tortonian and ends during the early

Quaternary. It corresponds to a compressive phase, with possible local occurrence of volcanism and a strike-slip component. The second phase starts clearly after 1.8 Ma and continue today. It corresponds to a strike-slip phase with an important extensive component. Both phases evidence the overall oblique convergence and important control of deep structures, which we will detail thereafter.

### 4.1.   Mio-Pliocene to Early Quaternary strain partitioning

The first tectonic phase is associated to Mio-Pliocene to Early Quaternary. The overall geometry of the SAR deformation shows a development of imbricated folds, distributed throughout a left-lateral shear zone, which is the result of oblique convergence and strain partitioning. Northward and southward dipping conjugated thrust faults accommodate the shortening at north and south flanks of the SAR (Fig. 6). The change of stacking pattern of the Pliocene deposits along the folds suggests a diachronous growth during the Pliocene with lateral variation of the uplift rates (Fig. 6 and 7). We interpret the geometry of



the fold as an evidence for a syn-folding sinistral shear (Fig. 10). This implies anti-clock wise rotations of the folds during the shortening. Such mechanism has been demonstrated along the Palomares fault in the Betic belt (Jonk et al., 2002). The aggradation of contourites at the foot of the SAR indicates a relative quiescence of the folding during the Quaternary 2.6 Ma (Juan et al., 2016).

Mio-Pliocene shortening is locally contemporaneous of volcanism. The lateral continuity of the highly reflective facies from
west to east suggests that the Small Al-Idrissi volcano could be equivalent of the Big Al-Idrissi volcano, but formed at a deeper bathymetry or vertically offset by local extension at younger times (Fig. 8 and 10). This highly reflective material triggers an acoustic masking of the reflections below (Fig. 9), as observed in debris avalanche deposits linked to recent volcanism in the Antilles Arc (Le Friant et al., 2002, 2009). The intercalation of this volcanic material toward the top of the Pl1 unit indicates that the Small Al-Idrissi Volcano and could be older than 4.5 Ma but younger than 5.33 Ma (Fig. 9). The NE-SW distribution
of the volcanic material suggests a syn-folding infill of the syncline axis (Fig. 10). Volcano-clastic deposits younger than the Messinian salinity crisis can be contemporaneous to the volcanism activity occurring to the North of the Alboran Ridge: the Djibouti Bank basalts, Ibn Batouta Seamount gabbro and the Macizo Volcanico de la Polacra syeno-diorites, dated between 5 Ma and 4.4 Ma (Duggen et al., 2008). This volcanism generally shows a high K content and occurs above thinned continental lithosphere (Duggen et al., 2008). It could be the produce of MCS decompression partial melting as proposed by Sternai et al.,
305   (2017).

Offshore tectonics structures in the Alboran Basin connects onshore in the Rif. The Nekor fault and the Jebha Fault accommodate a distributed deformation onshore and the transpressive deformation recorded around the SAR (Fig. 15a). The Jebha fault is in continuity with the SAR (Fig. 1 and 15), but stopped its activity before the Pliocene (Benmakhlouf et al., 2012). The passive infilling of paleo-ria indicates relatively low vertical motion (Romagny et al., 2014). After 3.8 Ma, a
transition from compression to radial extension (Benmakhlouf et al., 2012) causes NE-SW normal faulting in this region and produced a tectonic tilting of the Moroccan margin (Fig. 15b)(Romagny et al., 2014). Toward the south-east part of the Rif, the Nekor fault has acted as a transpressive fault zone accommodating the shortening (Ait Brahim et al., 2002; Aït Brahim and Chotin, 1990). The offshore extensive faults prolonging the Nekor fault are sealed offshore by Pliocene deposits and were inverted as blind thrust faults during the Plio-Quaternary (Watts et al., 1993). In the external Rif, interpretations of 2D seismic
reflection lines suggest a Late-Miocene tectonic transition from thin-skin to thick-skin deformation, linked to the inversion of the Mesozoic extensional structures (Capella et al. 2016). This thin-skin to thick-skin transition is recorded from Tortonian-early Messinian to Pliocene times, and causes the uplift of intramountainous basins around the Nekor fault (Fig. 15a)(Capella et al., 2016).

We propose that during the Pliocene the deformation partitioning recorded in the Rif and in the Alboran Basin evidences
oblique inversion of Miocene structures and strongly suggest that the deformation progressively switches from left-lateral transpressive to compressive (Fig. 15a and 15b). Basement strike-slip faults, oblique to the convergence direction, distribute the deformation into a rhombic structure of sigmoid fold and thrust. The distribution of the deformation into left-lateral motion and shortening reflects the onset of an oblique direction of shortening relatively from NE—SW basement faults (i.e. between



the Nekor and Jebha Fault and the Alboran Ridge). The left-lateral shear component of the deformation of the Alboran domain

implicates vertical axis rotation of the basement faults (Fig. 15a to 15c), as demonstrated elsewhere (Koyi et al., 2016; Tadayon et al., 2018). Vertical axis rotations favor a progressive change from transpressive to more purely compressive.

The development of oblique faults and thickness variation in the sedimentary cover, which results in non-cylindrical thrust wedges, lateral escape of frontal thrust sheets and vertical axis block rotations demonstrate the influence of a viscous layer at the base of the sedimentary covers, as demonstrated from analogue modelling (Storti et al., 2007). In the SAR, such a weak

layer can correspond to the early-Miocene under-compacted shales at the bottom of the sedimentary covers (Soto et al., 2008, 2012). Such weak layer can explain why the deformation significantly distributed in the SAR, whereas is appears to be more localized in the NAR. In contrast to the SAR region, in the NAR, the recorded uplift increases through time until it reaches a maximum around 2.45 Ma, with a clear pop-up structure (Martínez-García et al., 2017). It suggests that the AIF progressively decouples the deformation between the NAR and the SAR from 2.6 – 2.45 Ma.

### 4.2. Quaternary to present-day strain partitioning

#### 4.2.1. Evidences of Quaternary subsidence

The depth of the offlap breaks and the geometry of the shelves indicates a clear tectonic subsidence. This subsidence is contemporaneous with the northward tilting of the margin (Ammar et al., 2007). During the Quaternary, the Moroccan shelf underwent a local transgression and flooding characterized by the building of transgressive wedges on top of a prograding

clinoforms (Fig. 7 and 12). The retrogradation of the shoreline starts between 1.8 Ma and 1.12 Ma in the Nekor Basin and in the Big Al-Idrissi volcano (Fig. 7 and 12). Along the Big Al-Idrissi volcano, the depths of the offlap breaks are significantly lower than the maximum depths reached by the sea-level falls at Gibraltar during the Quaternary (Fig. 6 and 7) (Rohling et al. 2014), proving the tectonic subsidence.

A set of N-S normal/strike-slip faults controls the transgression of the shelf (Fig. 8 and 12). The onshore Trougout and Boudinar

faults, as well as the offshore normal faults affecting the Big Al-Idrissi volcano controls the subsidence and the segmentation of the onshore Moroccan margin. The Boudinar fault offsets early Pliocene marine sediments in the Boudinar Basin (Fig. 2b and 9) (Galindo-Zaldívar et al., 2015, 2018; Poujol et al., 2014). This set of normal faults represent an almost N-S fault network of *en-echelon* right-stepping normal faults (Fig. 10 and 15). At present day, the activity of N-S normal faults is unclear. Focal mechanism and microstructural studies demonstrate that this fault network is likely to be active with a sinistral component

(Fig. 4) (Poujol et al. 2014).

In the eastern part of the SAR, 145° normal faults are active with an orientation similar to the N140° normal faults accommodating the late-Pleistocene extension in the Nekor Basin (Fig. 10, 11 and 14) (Lafosse et al., 2017). From the local direction of the maximum horizontal stress field and from focal mechanisms (Fig. 2b and 4) (Neres et al., 2016), the fault zone is transtensive with a right-lateral motion. Thus, the 145° fault zone can be regarded as an antithetic or extensional structure



accommodating the present-day left-lateral motion along the AIF, or extensional structures related to the southern fault tip in the horsetail splay (Fig. 15c).

Therefore, the basin architecture demonstrates a spatio-temporal evolution of the AIF from 2.6 - 2.45 Ma to present day. The renewed uplift of the NAR around 2.6 -2.45 Ma (Martínez-García et al., 2013) is likely to be linked to the start of the activity along the AIF. The beginning of the transgression of the shelf around the Big Al-Idrissi volcano and the Nekor Basin is

approximately synchronous of the last shortening event along the NAR (1.8 to 1.12 Ma ; Fig. 8 and 12). In the Nekor Basin, the deformation localizes progressively along the strike-slip Boussekkour-Bokoya fault after 0.8 Ma (Lafosse et al., 2017), denoting a progressive localization of the deformation along the AIF and a westward migration of the deformation as proposed in Galindo-Zaldivar et al., 2018 and Lafosse et al., 2017 (Fig. 15c).

### 4.2.2.  Quaternary evolution and localization of the Al-Idrissi fault zone

We propose that the beginning of the subsidence of the shelfs near the AIF dates the activation of the shear zone. During the Quaternary and until present-day, the AIF is a regional left-lateral shear zone decoupling the deformation in the SAR and the NAR (Fig. 9). The AIF acts as a transfer fault accommodating the shortening north of the Alboran Ridge in the Djibouti Bank Area (Estrada et al., 2018) and the Rifian extrusion along the Nekor fault. The AIF has progressively propagated southward, activating the N-S right-stepping normal fault linking from the AIF to Boudinar and Nekor Basins during the Quaternary (Fig.

15b). Since *ca*. 1.8 Ma-1.12 Ma, the transgression of the shelf of the Big Al-Idrissi volcano and the inception of the subsidence of the Nekor Basin indicate a localization of the deformation on a releasing bend activating N-S faults. The compressive bend in the northern part of the AIF affect the seafloor and is activated during recent seismic crisis (Buforn et al., 2017; Galindo-Zaldívar et al., 2015). West of the AIF, the N145° normal-strike-slip faults are likely recent, accommodating the sinistral motion along the AIF and the Bokoya and Boussekkour Faults. The apparent low lateral offset and the localization of the

deformation on the strike-slip Boussekkour-Bokoya fault zone after 0.8 Ma suggests that the localization of the deformation along the AIF is a recent feature (Fig. 15c) (Lafosse et al. 2017). In this context, the normal faults are equivalent to antithetic faults within a horsetail splay, that connect to the Trougout Fault and the Nekor faults (Fig. 15b). Such structures grow probably through a mechanism of relay ramp, like the one proposed in other strike-slip contexts such as the Paleogene Bowey Basin (Peacock and Sanderson, 1995).

The localization of the deformation along the AIF could be controlled by a Miocene pre-existing structure as proposed in Martínez-García et al., (2017). At a crustal scale, geophysical studies show a strong crustal thickness variation at the Al Hoceima region (Diaz et al., 2016) which can contribute to the localization of the deformation. The contrasted crustal thickness origins either in a Miocene oblique collision (Booth-Rea et al., 2012), lower crust doming during the Miocene transtension (Le Pourhiet et al., 2014), or lower crust removal associated to delamination processes (Bezada et al., 2014; Petit et al., 2015). The

localization on crustal heterogeneity has been evidenced in numerical models for example in cratonic lithosphere (Burov et al., 1998). It follows that the localization of deformation along the AIF evidences the strong control of the crustal heterogeneities. This localization occurred in a short period from 1.8Ma to present-day.



### 4.3.      Transpressive or transtensive deformation?

Changes of tectonic style in the Alboran Basin have been related to changes of direction of far-field forces (Martínez-García
et al., 2013). However, an alternative explanation can be proposed. Since 5 Ma, the direction of Africa Eurasia convergence
remains constant (DeMets et al. 2015). In an absolute reference frame, the direction of convergence between Africa and Eurasia
is NNE-SSW, producing 15km of shortening since 8Ma (Spakman et al. 2018). From GPS measurements and from present-
day stress and strain modelling, the Alboran tectonic domain can be considered as domain undergoing a clockwise rotation of
1.17°/Ma (Palano et al., 2013, 2015). This value has the same order of magnitude to the domino model from Meghraoui and
Pondrelli (2013) of a tectonic block undergoing a long term clockwise rotation of 2.24°/Ma to 3.9°/Ma. In the Alboran Basin,
the TASZ must rotate as well to accommodate the convergence and block rotation (Fig. 15). Since 5.33 Ma, estimates the
rotation of the Alboran tectonic domain are around 13°-20° clockwise (Fig. 15)(Meghraoui and Pondrelli, 2013). The obliquity
angle between major fault planes of the Alboran ridge and the direction of far-field forces (i.e. Africa-Eurasia convergence)
decreases progressively between the MSC and the late Pliocene (DeMets et al., 2015). This leads to the present-day more
orthogonal compression along the NAR (Fig. 2b and 15)(Cunha et al., 2012; Neres et al., 2016) and the inversion of the Alboran
Basin and the Algerian Margin (Derder et al., 2013; Hamai et al., 2015; Martínez-García et al., 2017).

Therefore, the evolution of the deformation in the Alboran Ridge represents the expected evolution of transpressive structures
under a constant oblique shortening and indentation of the African lithosphere (Fig. 15). Block rotations, transpressive folds
propagation and their anti-clock wise rotations, followed by transtensive deformation during the inception of the AIF, represent
successive steps within the tectonic inversion of the Alboran Domain since 8Ma. From 8 Ma to present day, the oblique
shortening has been distributed between the SAR and the Nekor fault (Fig. 15)(Booth-Rea et al., 2012; Do Couto et al., 2016).
The late Miocene-early Pliocene period in the Rif Belt matches the uplift of the Miocene intramountainous basin along the
Nekor fault under a transpressional tectonic regime (Fig. 15a). The uplift of those basins corresponds to the change from thin-
skin to thick-skin deformation in the external Rif during the inversion of the deep Mesozoic extensive structures (Capella et
al., 2016; Martínez-García et al., 2017) and during the transpressive deformation in the Temsamane units (Fig. 1)(Booth-Rea
et al., 2012). It suggests a progressive mechanical coupling between the African Margin and the Alboran Domain, locking the
Nekor fault in its eastern segment (Fig. 15). Progressive vertical axis rotation associated to shortening of the Alboran domain
decreases of left lateral motion, and an increase of compressive deformation along the Alboran Ridge (Fig 15b). Eventually,
the deformation has localized on the AIF during the early Quaternary decoupling the deformation between the NAR and the
SAR with an incipient transtensive mode (Fig. 15c). It induces a change of strain partitioning along the TASZ illustrated by
the transition from a Pliocene left-lateral shearing and folding of the SAR to a transtensive Quaternary deformation localized
on the AIF and the Nekor Basin (Fig. 15).

In this framework, normal strike-slip behaviour observed to the north of the NAR (Fig. 1)(Giaconia et al., 2015; Gràcia et al.,
2012; Grevemeyer et al., 2015; Palomino et al., 2011) goes a step further in the sense of an indentation of the Africa plate into
the Alboran tectonic domain (Fig. 2)(Estrada et al., 2018; Palano et al., 2015). This indentation is accommodated through the





left-lateral AIF and the right-lateral Yusuf fault zone (Fig. 1 and 15) in a similar way than the Palomares fault zone transferring the orthogonal shortening of the Iberian margin toward the Carboneras fault zone and the Central Alboran Sea (Estrada et al., 2018; Giaconia et al., 2015). In the SAR and the Nekor Basin, the present-day deformation under transtensive regime (NNW-SSE to N-S extensive network and NNE-SSW strike-slip faults; Fig. 4 and 15) is limited to the east by the Al-Idrissi fault. The

deformation in the NAR is on the contrary clearly compressive (Estrada et al., 2018; Martínez-García et al., 2017) and the geodetic data indicates similar displacements in the EAB and in the Rifian units east of the Boudinar Basin (Koulali et al., 2011; Vernant et al., 2010). Such difference of behavior suggests that the AIF may represent the present-day plate boundary between Africa and Alboran Domain.

## 5.    Conclusion

This study focuses on the tectonic evolution of the southern margin of the Alboran Sea during the Plio-Quaternary period, and particularly the distinct structural evolutions and interactions of the AIF and the AR, and the mechanisms associated to their formation. The analysis of the seismic stratigraphy and the comparison between onshore and offshore tectonic structures leads to the following tectonic framework:

(1)    The TASZ localizes the deformation between the Miocene and the end of the Pliocene in particular in the Alboran
Ridge and the SAR transpressional zones. Its inherited orientation favors a strike-slip movement during its oblique shortening. The rhombic folded structures of the SAR illustrate an important left-lateral displacement still active during the Pliocene. Consequently, during the Pliocene, the SAR accommodates the strain partitioning between left lateral and shortening.

(2)    Under the indentation of African lithosphere, vertical axis block rotations which lead to a progressive compression
on the Alboran Ridge and a younger activation under left lateral transtension along the AIF. The subsidence of both the Nekor Basin and the Big Al-Idrissi volcano demonstrate the start of the transtensive deformation between 1.8 Ma and 1.12.

(3)    The SAR undergoes transpression whereas further east tectonic inversion of the Algerian and Iberian margin occurs. The area between the SAR and the Nekor fault is progressively extruded, whereas east of the Al-Idrissi fault, the
African lithosphere indents the Alboran tectonic domain. The AIF transfers this indentation to the Nekor Basin, which accommodates the present-day westward extrusion of the Rif and represent an incipient plate boundary since 1.8 Ma.

Our findings demonstrate that at the scale of a basin, strike-slip shear zones evolve in response to far field forces but also in response to the local evolution of the fracture zone. This evolution can be fast and be achieved in a less than 2 Ma. Further researches are needed to better understand what drive the timing and the evolution of such large scale-strike slip structures.





## 6. Author contribution

Manfred Lafosse wrote the paper and conducted the study.

Elia d'Acremont and Christian Gorini lead the oceanic surveys MARLBORO-1, 2 and SARAS. The contributed to study and to the redaction of the present paper.

Alain Rabaute contributed to the data acquisition and processing and to the redaction of the present paper.

Ferran Estrada contributed to the data acquisition and processing.

Martin Jollivet-Castelot contributed to the stratigraphic interpretation.

Juan Tomas Vazquez contributed to the data acquisition and processing.

Jesus Galindo-Zaldivar and Gemma Ercilla contributed to the data acquisition and processing. They are the PI of the INCRISIS survey. Gemma Ercilla also contributed to the stratigraphic correlations and interpretations.

Belen Alonso contributed to the stratigraphic correlations and interpretations.

Abdellah Ammar contributed to the data acquisition.

## 7. Acknowledgement

We thank the members of the SARAS and Marlboro cruises in 2011 and 2012. This work was funded by the French program Actions Marges, the EUROFLEETS program (FP7/2007-2013; n°228344), project FICTS-2011-03-01. The French program ANR- 17-CE03-0004 also supported this work. Seismic reflection data were processed using the Seismic UNIX SU and Geovecteur software. The processed seismic data were interpreted using Kingdom IHS Suite©software. This work also benefited from the Fauces Project (Ref CTM2015-65461-C2-R; MINCIU/FEDER) financed by "Ministerio de Economía y Competitividad y al Fondo Europeo de Desarrollo Regional" (FEDER).

## 8. Competing interests

"The authors declare that they have no conflict of interest."

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





## 10. Figures

Figure 1: Topographic map and major structural units of the Alboran region. Structural units in the studied area modified from Chalouan et al., (2008); Comas et al., (1999); Leblanc and Olivier, (1984); Romagny et al., (2014). The Trans Alboran Shear Zone (TASZ) indicates the motion inferred for the Late-Miocene – Pliocene period. The red fault is the present-day active Al-Idrissi fault AC, Alboran Channel; CF, Carboneras Fault, EAB; East Alboran Basin; AIF, Al-Idrissi Fault; JF, Jebha fault; NF, Nekor Fault; SAB, South Alboran Basin; SAR, South Alboran Ridge; NAR, North Alboran Ridge; YF, Yusuf Fault; WAB, West Alboran Basin. Inset: Hypotheses of plate boundaries between a Alboran tectonic domain and the African plate from Nocquet, (2012).




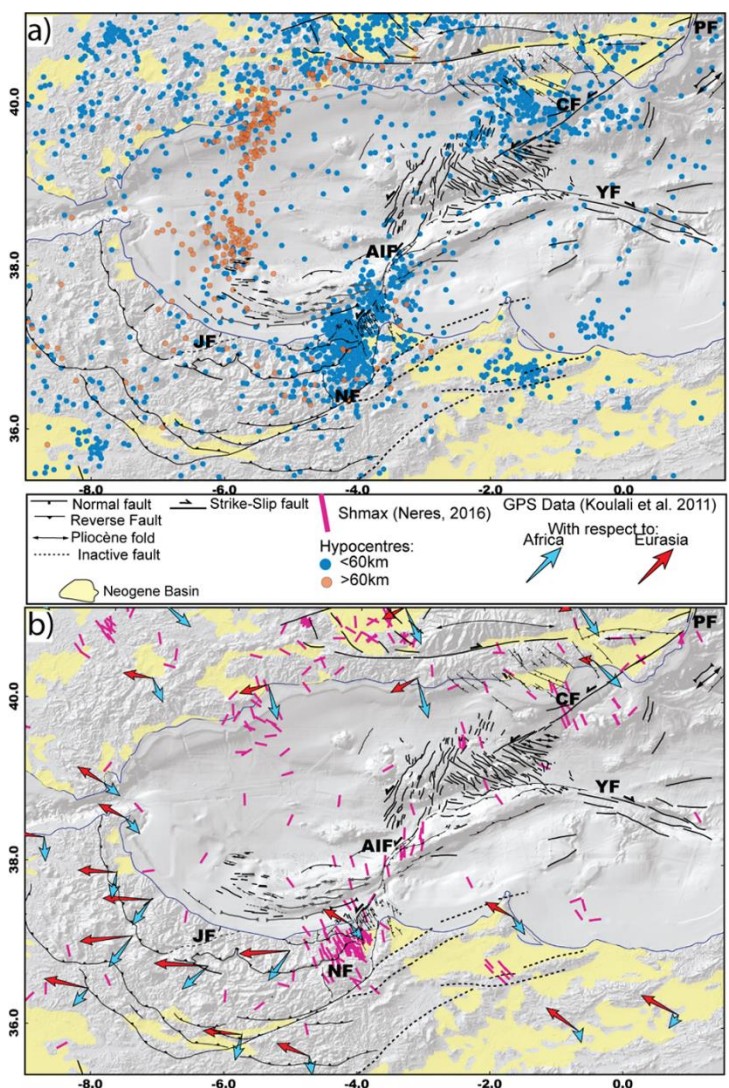


**Figure 2 : Maps showing the distribution of the seismicity along the Neogene tectonic structures in the Alboran Sea. a) Neotectonic map of the Alboran region modified from and from this study. Seismicity from IGN catalogue 1970-2017 (http://www.ign.es/), only earthquakes with Mw >= 3 and depth >=2 km are figured. b) GPS data from Koulali et al., (2011) and Sh$_{max}$ from Neres et al., (2016). See figure 1 for scale. CF, Carboneras fault; PF, Palomares fault; YF, Yusuf fault; NF, Nekor fault; AIF, Al-Idrissi fault zone.**






**Figure 3.** Bathymetry of the studied area showing the main morpho-structural features of the studied area. Dashed black lines, position of the seismic lines used in the study. MTD, Mass Transport Deposits; WAB, West Alboran Basin; SAB, South Alboran Basin; BF: Boussekkour Fault; Bof, Bokoya Fault; BiF, Boudinar Fault; NF, Nekor Fault; AIF; Al-Idrissi Fault zone; Trougout fault; AjF, Adjir-Imzouren Fault.


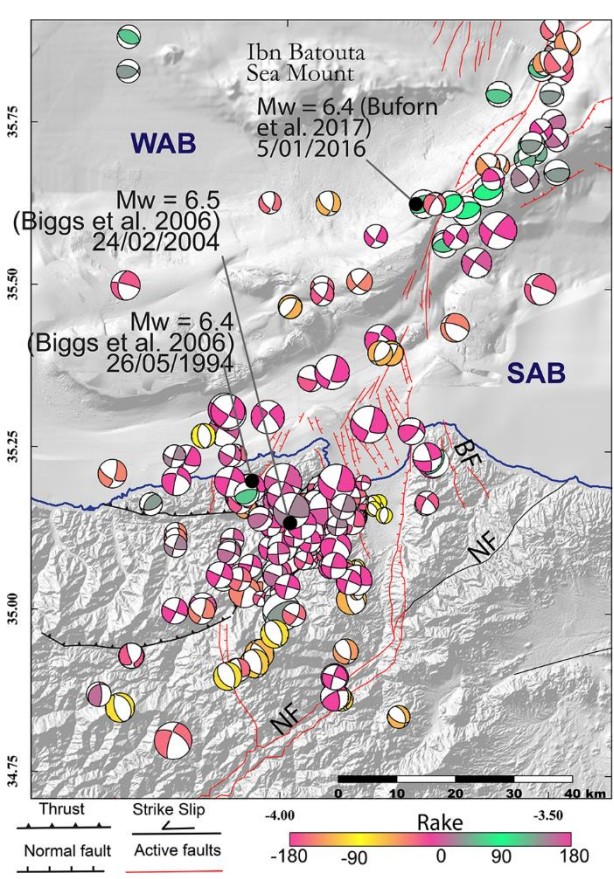

**Figure 4 :** **Map of the distribution of the present-day deformation showing strike slip and compressive deformation along the northern part of the studied area and extensive and strike-slip structures along the southern part. Focal mechanism till 2014 period from the compilation of Custódio et al.(2016) and for the year 2016 from GCMT project ; http://www.globalcmt.org/; Dziewonski et al., 1981; Ekström et al., 2012). Size of the focal mechanisms corresponds to the magnitude values (from Mw= 2.3 to 6.4). Structural data compiled from Ballesteros et al., (2008); Biggs et al., ( 2006); Buforn et al., (2017); Chalouan et al., (1997); Lafosse et al., (2017) and Martínez-García et al., (2011). BF, Boudinar Fault, WAB, West Alboran Basin; SAB, South Alboran Basin; NF, Nekor fault.**



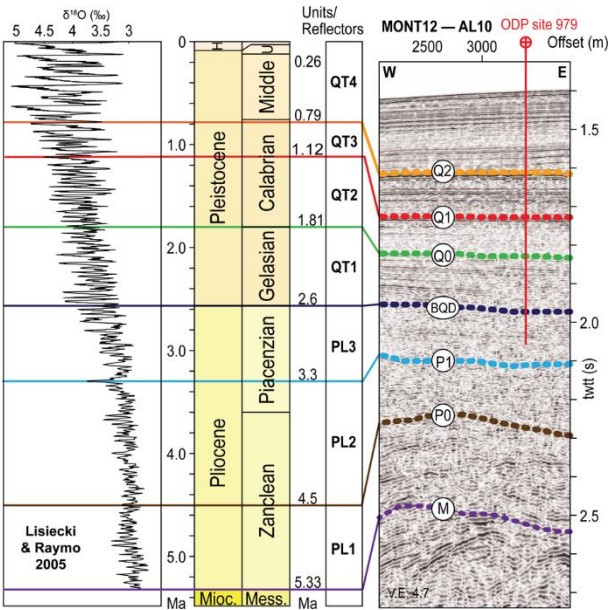

**Figure 5 : Well log correlation to the seismic section, seismic line crossing the location of the ODP 979 site, vertical stacking of the Pliocene and Quaternary units and available δ18O curve from Lisiecki and Raymo (2005). The colors of the stratigraphic surfaces are the same as in the following seismic lines.**






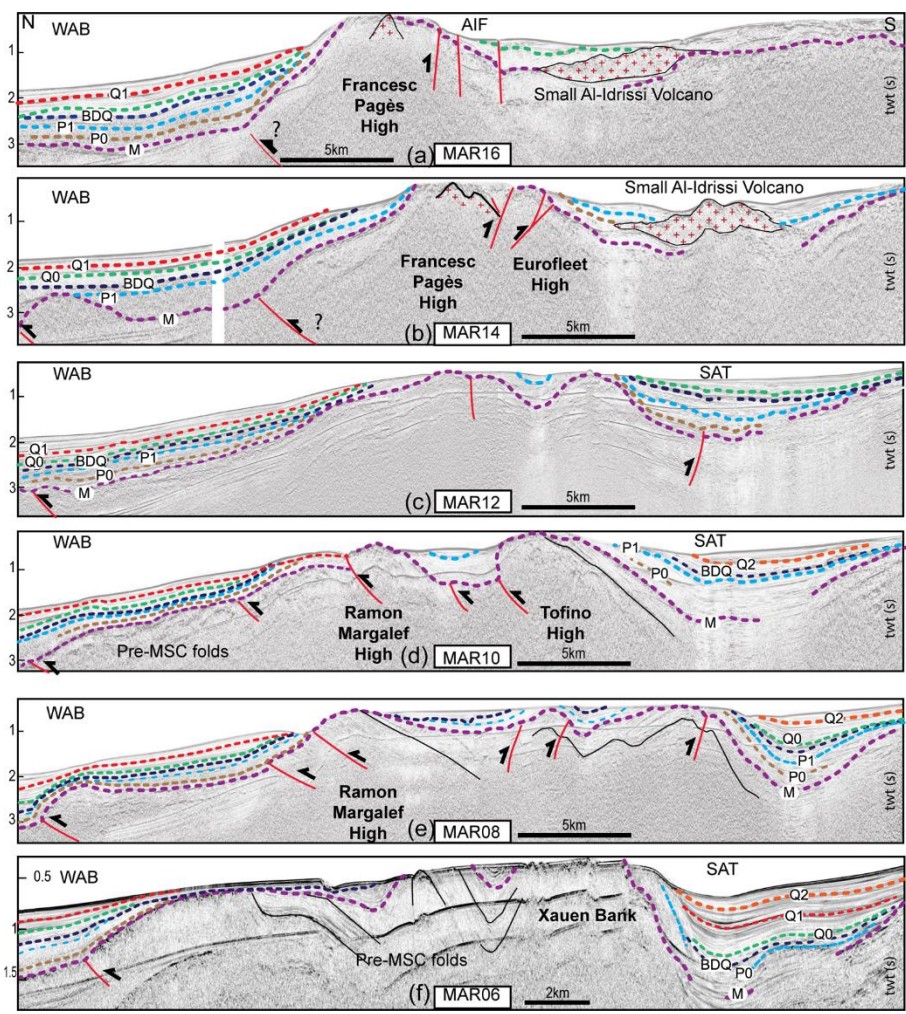

**Figure 6: Multichannel seismic records showing the Plio-Quaternary stratigraphy and structural features. Dashed and colored line are the stratigraphic surface defined in figure 5. Black, reflectors, pre-MSC reflectors. The seismic section (a) to (f) are ordered from east to west. WAB, South Alboran Basin; SAT, South Alboran Trough; AIF, Al-Idrissi Fault zone.**




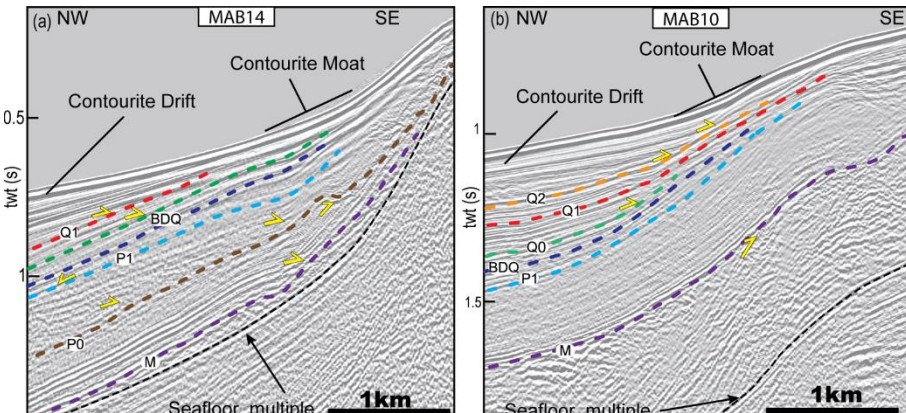

**Figure 7: Seismic unconformities at the foot slope of the northern flank of the South Alboran Ridge. a) Seismic line at the foot of the Francesc Pagès bank. b) Seismic line at the foot of the Ramon Margalef high. The seismic lines show the diachronism of the deformation affecting the SAR during the Pliocene. After 2.6 Ma, the moats of the contourites pinch at the feet of the folds.**






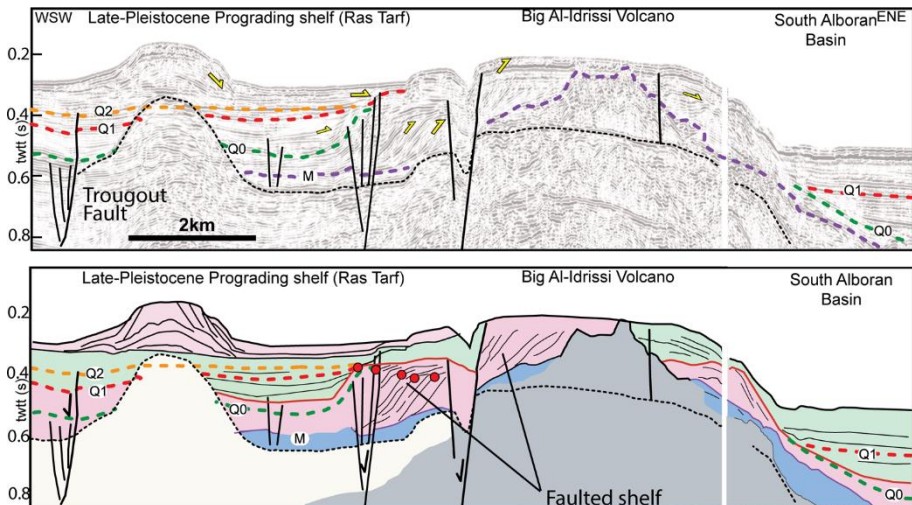

**Figure 8: Multichannel seismic profile showing the transgression of marine sediment (in green) over the prograding shelf to the edges of the Big Al-Idrissi volcano (in pink) crossing the Ras Tarf Promontory and the Big Al-Idrissi Volcano. Dashed black reflector, multiple of the seafloor. Red points, offlap break (Paleo-shore line) marking the concave up trajectories of the offlap breaks and progradation of the shelf and a first transgression before 1.81Ma. The red surface is a maximum regressive surface in the sense of Catuneanu et al. (2011). The seismic line shows the transgression of marine sediment (in green) over the Pliocene to Quaternary prograding shelf to the edges of the Big Al-Idrissi volcano (in pink). Older depositional units are colored in blue and the acoustic basement in grey.**



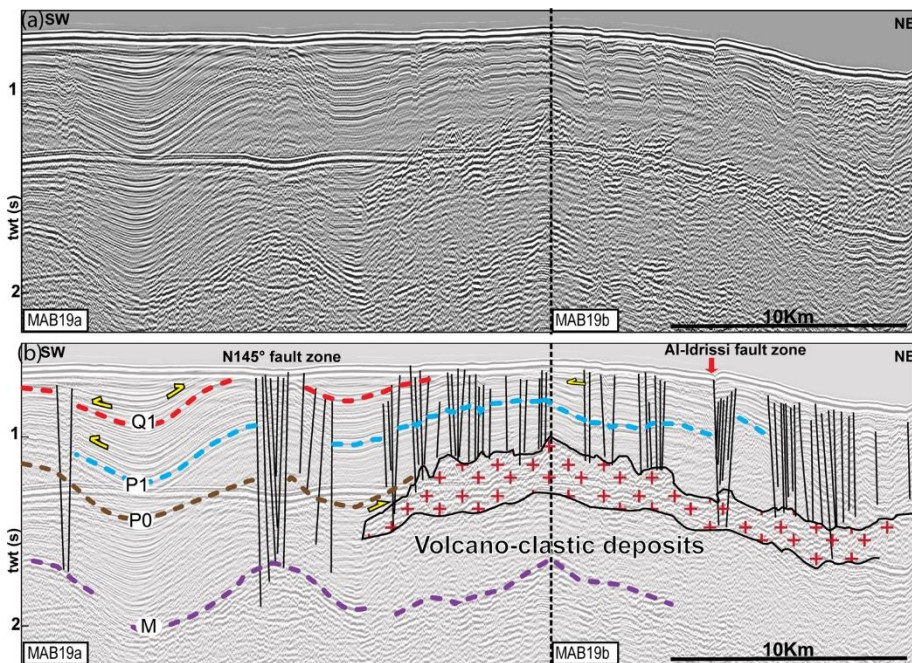


**Figure 9: Multichannel seismic profile showing seismic stratigraphy and the main structural elements along a portion the South Alboran Trough located between N145° striking faults and the AIF. Line track on figure 3. (a) Raw seismic line (b) Interpreted seismic line. Red-crosses in b) figure a seismic body made of poorly continuous high-amplitude reflectors interpreted as volcano-clastic deposits.**




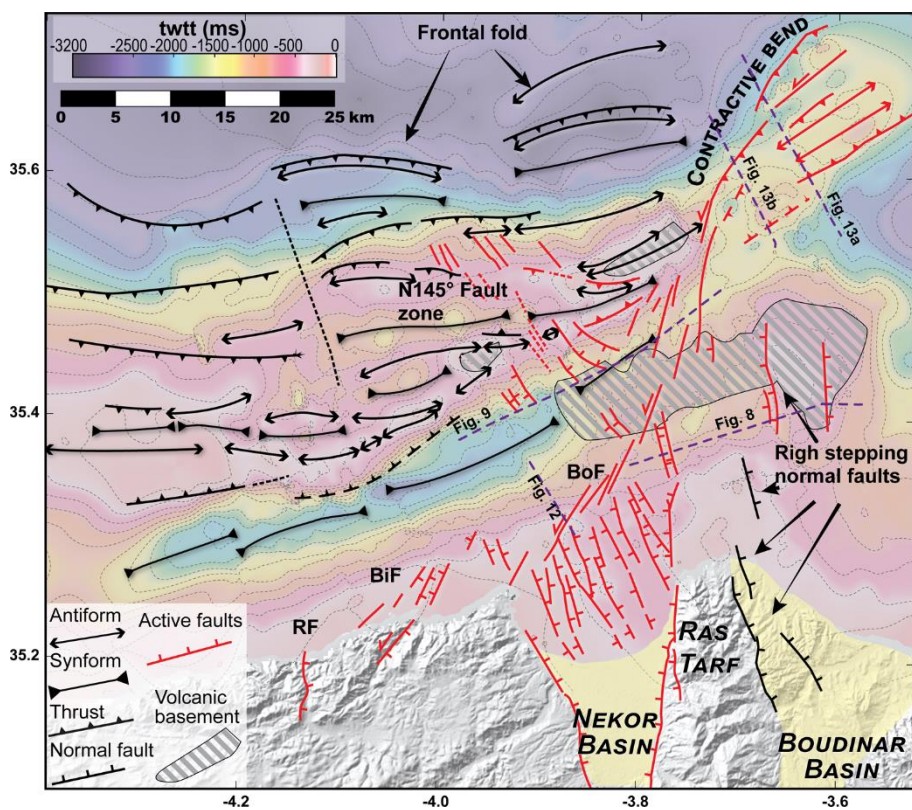

**Figure 10: Structural map of Plio-Quaternary faults and folds overlying the map of depths of the Messinian unconformity. Active faults corresponds to the faults affecting the seafloor. BF: Boussekkour Fault; Bof, Bokoya Fault; RF, Rouadi Fault.**





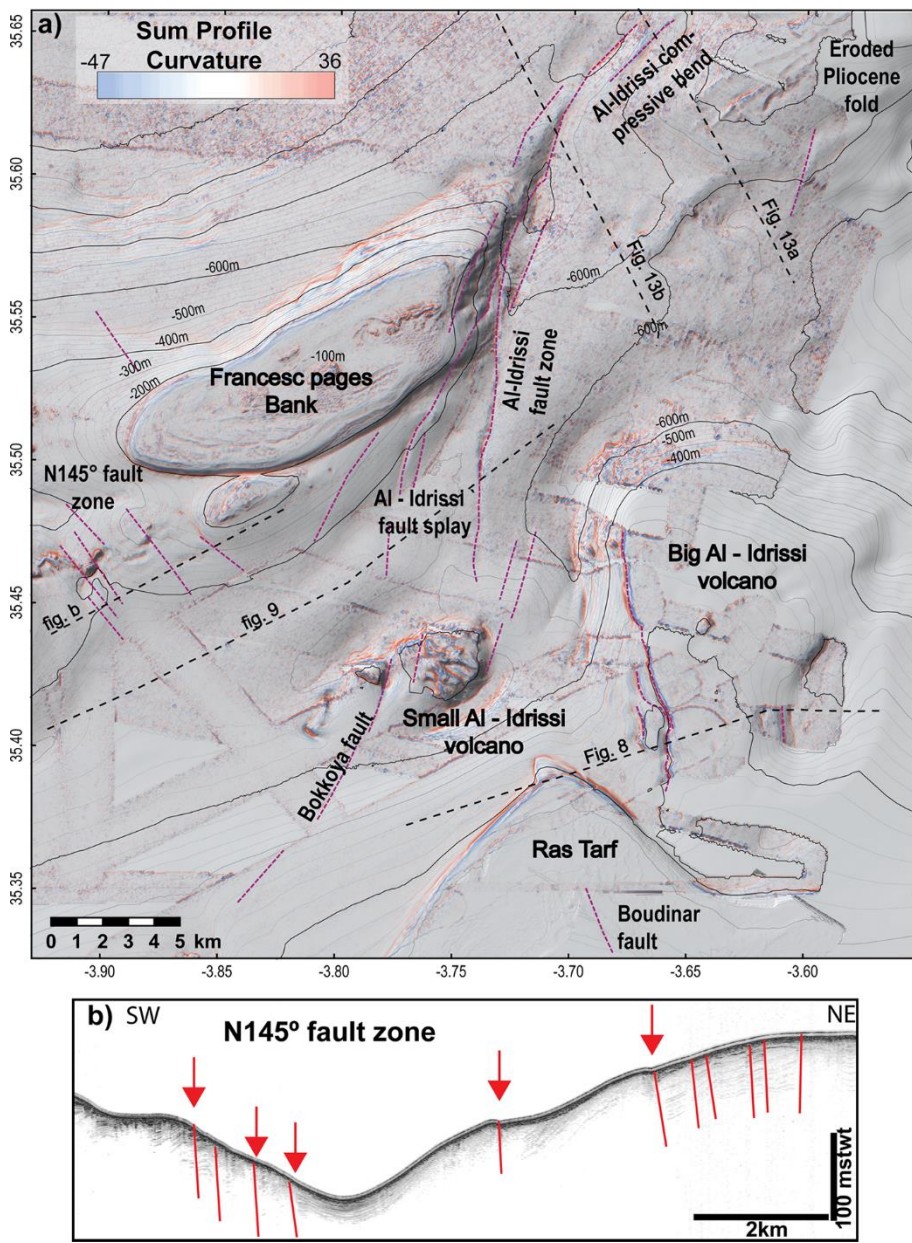


**Figure 11: Active structures around the roughly NNE-SSW AIF and adjacent submarine highs. The AIF bends to the North where it follows the trends of the NAR. High values of curvatures in the Francesc Pagès Bank and in the Northeast corner of the map underline the linear features at the seafloor, which corresponds to the truncated Miocene-Pliocene layers. Extreme positive values in red represent concave topography at the seafloor; extreme negative values in blue represent convex topography. a) profile**

**curvature map textured above the shaded bathymetry; dashed purple lines, fault tracks at the seafloor; dashed black line, position of the seismic line in b and in figure 8, 9 and 13. b) TOPAS profile showing active N145° normal faults. Red lines, active faults; red arrows, position of the fault traces in (a).**







**Figure 12: SPARKER seismic line showing the transgression of marine sediment (in green) over the prograding shelf of the Nekor**
**Basin (in pink). Oldest depositional units (Pliocene) are colored in blue and the acoustic basement in grey. The Maximum Regressive**
**Surface (MRS) is in red.**



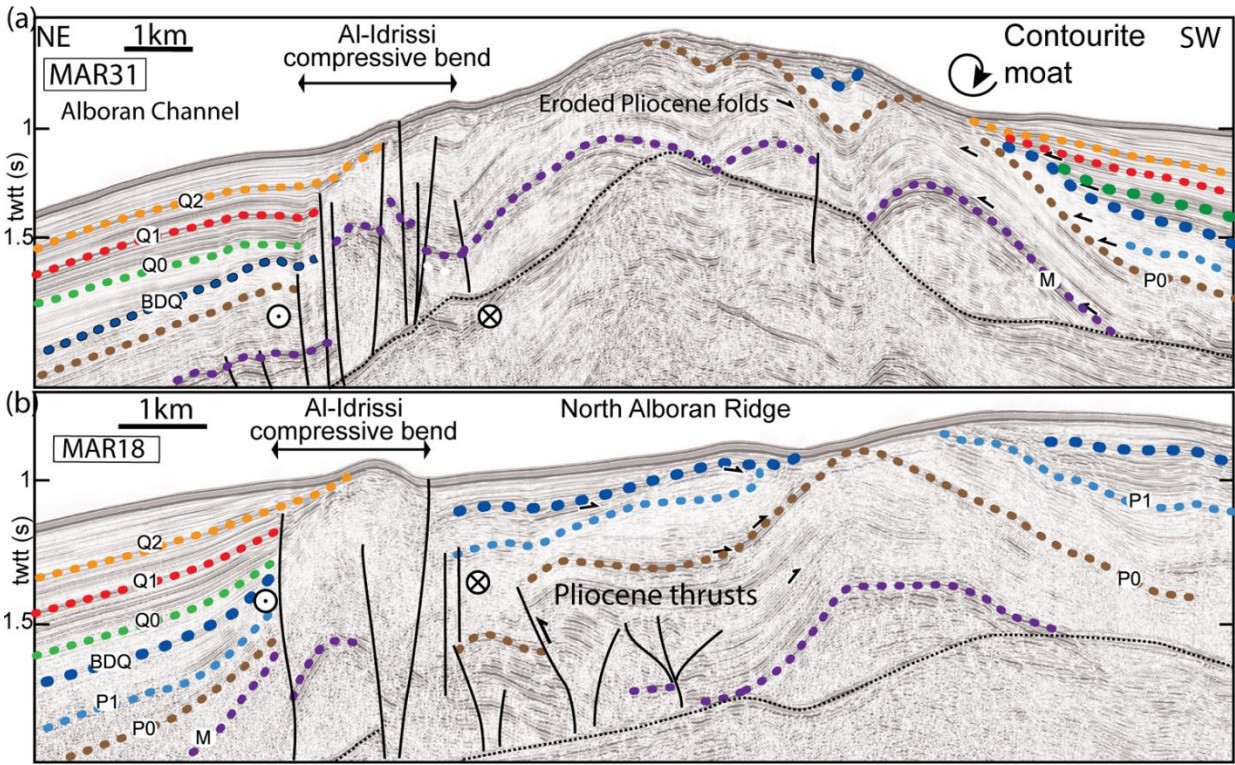

**Figure 13: Multichannel seismic lines across the compressive bend of the Al-Idrissi fault zone showing lateral evolution of the tectonic structures in North Alboran Ridge and in the compressive bend. a) The Al-Idrissi fault zone is a positive flower structure at the northern front of the Alboran Ridge. b) The Al-Idrissi fault zone is a positive flower structure distinct from the Pliocene thrusts and folds affecting the Alboran Ridge.**





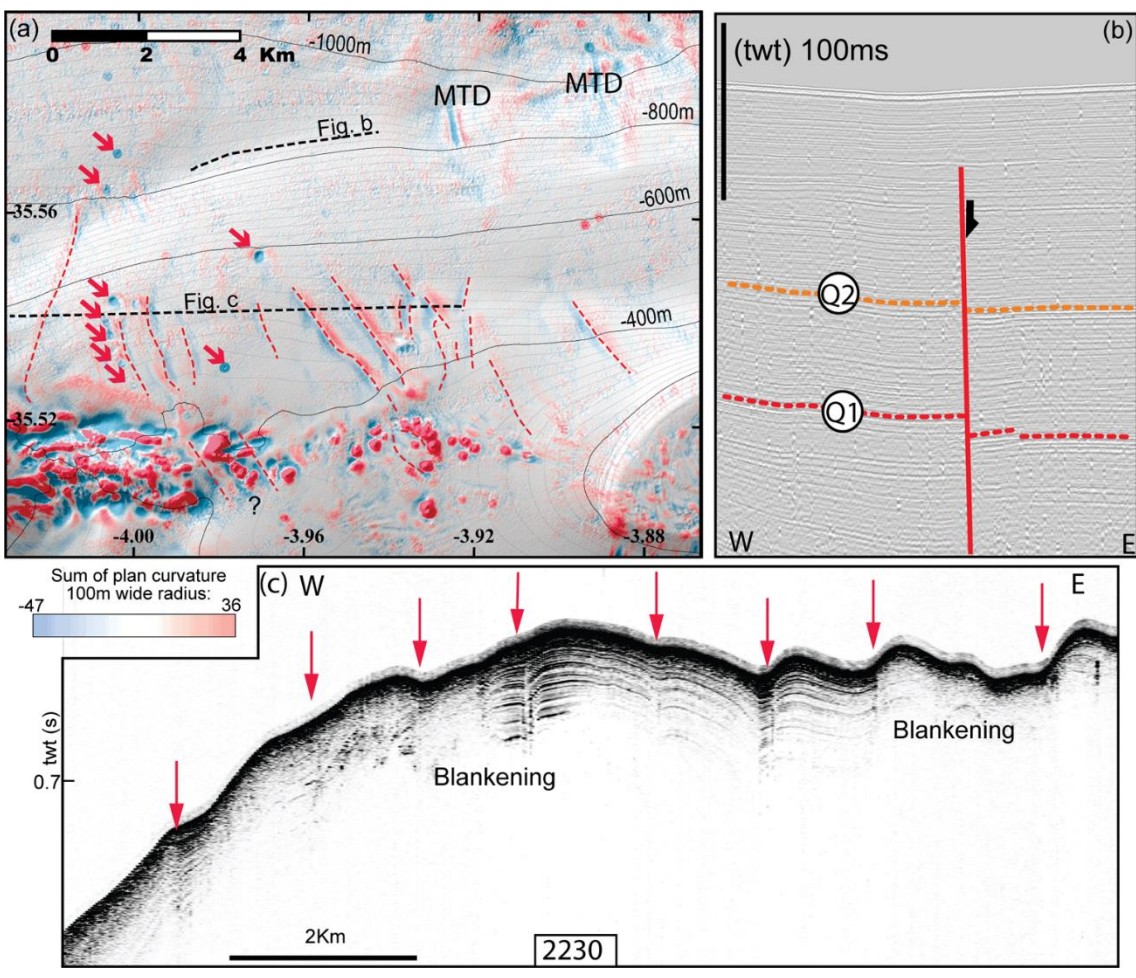

**Figure 14: Active structures affecting the northern flank of Francesc Pagès and Ramon Margalef highs. a) plan curvature map overlying the shaded bathymetry; red arrows pockmarks on the seafloor; dashed black lines, seismic lines in Fig. b and c; dashed red lines, positions of the fault track. b) SPARKER seismic reflection line showing the northward continuity of N145° fault (redline). c) TOPAS seismic line showing the subsurface of the seafloor. Red arrows, position of the faults drawn in a).**









**Figure 15: Palinspastic maps of the SAR and the Rif from 5 Ma to the present-day are using 14 ° clockwise rotation of the Alboran tectonic domain from a) to c). Dashed blue line, approximate coastline; continuous blue line, present-day coastline; Dark yellow, Miocene-Pliocene onshore basins; light yellow, Pliocene and Quaternary onshore basins; grey patch, position of the slab remaining approximatively constant below the Alboran Basin during the Plio-Quaternary; left bottom corner of the maps, simplified drawing**

**figure the area between the SAR, the Nekor fault and the Yusuf fault. Thick grey arrows in (c) indicate the direction and relative amount of extrusion in the central Rif considering a fixed Eurasia. The shortening is accommodated though compressive structures in (a). The initiation of subsidence along the Big Al-Idrissi Volcano and the Moroccan shelf corresponds to (b) and the present – day partitioning of the deformation corresponds to (c). CR, central Rif, JF, Jebha Fault; NF, Nekor Fault; AIF, Al-Idrissi Fault zone.**