# Peer review of "Plio-Quaternary tectonic evolution of the southern margin of the Alboran Basin (Western Mediterranean)"

_Solid Earth, 2019_

## Short Comment (SC1) · 3 Sep 2019

In this paper the authors propose a Plio-Quaternary tectonic evolution of the southern margin of the Alboran Sea, mainly based on the interpretation of multichannel seismic reflection profiles and other relevant stratigraphic information and multibeam data. They identify at least two evolutionary phases for this area: (i) a first ,mostly compressive phase of Tortonian age, ending during the early Quaternary, with a remarkable development of imbricated folds and local occurrence of volcanism and strike-slip structures; (ii) a strike-slip phase with a significant extensional component, which started after 1.8 Ma; within this phase, an important role has been played by the Al-Idrissi fault

zone (AIF), which splits the Alboran ridge and, according to the author's interpretation, may represent the present-day plate boundary between Africa and the Alboran domain. As a general comment, the paper should be significantly reduced in length and better organized, especially in the data presentation: in the present form, it is quite difficult to follow the text because it imposes to flip from one figure to another. This because the location maps are distributed in several figures, and for this reason it is difficult to understand the key points presented by the authors. I suggest to present a single index map in which all the presented profiles are indicated in bold. In the paragraph Data, the authors should present the summary of the acquisition parameters used, and the data processing done (better in a dedicated Table) Another point which should be clarified is the relationship between the Quaternary subsidence and the strike-slip tectonics, as presented in paragraph 4.2.1. Indeed, it is not clear from the text. Is this part important in the general context of the paper? In addition, I do not see the importance of the question raised in the paragraph 4.3. It is quite obvious that changes in tectonic styles (and the consequent structural elements produced) are related to changes of direction of stress field. The authors should identify what are the most important structural elements derived from the interpretation, and propose a plausible mechanism, avoiding such a long discussion. Some of the seismic profiles cross the AIF. Apparently, in the presented data it does not appear as a regional, relevant strike-slip fault accommodating the oblique movement between the two domains. In the earthquake distribution map, the main clusters are located to the south, within the Nekor Basin, and only few events are located at the Alboran ridge. Have you an interpretation about this? Possibly we are facing not a single structure, but most probably a sequence of sub-vertical lineaments distributing the strain, or, in other words, a diffuse transfer zone. Most of the presented seismic profiles contain the interpretation superposed. This makes difficult to the reader to follow the interpretation and verify it goodness. For this reason, I suggest to present the uninterpreted and interpreted version of the lines (as an example, the profiles in Fig. 6). Finally, I suggest having the text corrected by a native English speaker.

Based on the above, I suggest a moderate revision for this paper.

Specific points: Line 41: takes control? Line 52-53: . . .around 8-7 Ma in the. . . Line 62: Dot after the ) Line 64: use the acronym WAB Line 66: delete "this" Line 77: . . .of the southwestern. . . Line 78: We analyze Line 84: write "is", instead of "corresponds to" Line 108 and 111: extensional Line 129: has triggered? Line 133: relative displacement? Please explain which plates are involved Line 160: multichannel seismic profiles Line 171: . . .interpretation to perform the. . . Line 178: why you use this acoustic velocity? Have you performed velocity analyses on the data? This should be clarified Line 183: sedimentary sequence Line 192: evidence Line 198: . . .to a 4-8 km wide conic. . . Line 199: This sentence is unclear. . .rewrite Line 212: unconformably Lines 219-220: conversion in depth based on what velocity? (see comment above) Line 225: this sentence is unclear. . . Line 241: this sentence is unclear. . . Line 247: ??? Line 254: . . .fault zone composed by. . . Line 290: why sinistral shear? Explain Line 292: ..Quaternary at. . . Line 295: equivalent volcano? This is unclear. . .re-phrase Line 304: . . .could be the product of MCS. . . (please explain the acronym and the meaning of this) Line 309: paleo-ria??? Line 325: implies Line 336: Evidence of. . .

Caption of Fig. 2: . . .modified from and from. . .??? Caption of Fig. 3: indicate NB, BB, . . . Fig 6: please present uninterpreted and interpreted profiles, at a larger scale! Caption of Fig. 8: there is a repletion of sentence (the seismic line shows.) Fig. 9: I do not see the location of this figure

---

## Referee Comment (RC1) · Jacques Déverchère (Referee) · 6 Sep 2019

This paper deals with the recent tectonic evolution of part of the southern margin of the Alboran Basin (and not the whole southern margin as wrongly stated in the title). Actually, the authors focus on the offshore southernmost part of the Trans-Alboran Shear Zone (TASZ) representing a broad area of deformation which is not as well documented as other areas in the Alboran domain. The interest of the work is to further document this area with new, high-quality seismic data of high to very high resolution and to better assess and/or discuss the reasons for the fast stress changes that occurred since Pliocene. As a whole, this contribution appears stimulating and rather

convincing and is worth to be published. However, several limitations appear in the way the authors reports previous studies and discuss their interpretations; furthermore, the bad organization of the figures and the poorly written English make the paper quite difficult to read. Actually, I have identified an important weakness: an important point supported by the authors is to suggest that besides the well-assessed processes of indentation in the central Alboran domain and extrusion of the Rif, the major change of tectonic style in the study area is related to a clockwise rotation of the Alboran tectonic domain instead of a change in the Eurasia/Africa plate convergence vector. The reality is that (1) uncertainties on kinematic data (DeMets at al., 2015) prevent from assessing any significant change in the obliquity angle or in slip direction in the study area since the Messinian, although Nubia-Eurasia angular velocities estimated from geodetic and geologic observations appear to differ significantly; (2) the block rotation model proposed by Meghraoui and Pondrelli (2013) is a large-scale model ("restraining bend") based on the assumption of a right-lateral deforming zone with large fault systems in the offshore domain: in my opinion, the fault geometry, sense of motion and continuity are far from being clearly assessed, preventing from concluding firmly that a block rotation model (either bookshelf or pinned) is responsible for the change in tectonic style. It results from (1) and (2) that the choice made by the authors is quite questionable, in my opinion. Other parameters such as, for instance, changes of body forces during crustal thickening of the South Alboran Ridge or even further south, or transpressive fold propagation (as suggested, lines 403-404), or also effects of propagation of the Al-Idrissi fault on strain distribution, should be included in the discussion on this challenging question. I have other comments below: - The introduction part is full of unclear statements (i.e., lines 79-80, 81-82, 129-130, etc…), so that facts are often confused and mixed with interpretations. I recommend to carefully check the reported assessments in order to clarify between what is established and what is an hypothesis. - The term "inversion" is systematically used throughout the paper, however it appears that not only inversion is actually occurring, but merely reactivation, or tectonic re-organization. I urge the authors to distinguish a true tectonic inversion of

structures (or of the margin) from other types of tectonic changes (for instance, from a strike-slip to an extensional strain change). I also urge the authors to better highlight the new features and facts they bring compared to previous studies instead of speculating too much on a possible cause of strain re-organization. For instance, the clear change of strain pattern (both in tectonic style but also in strain expression) shown on Figure 9 between Pliocene and Quaternary is an excellent example of strain superimposition at short time scale on a same geological structure and deserves for instance a comparison with the well-expressed NW-SE set of diffuse, secondary faults identified between the Carboneras fault and the north Alboran Ridge further north (Perea et al., Marine Geology, 399, 23-33, 2018). - The role of volcanism in the tectonic evolution is only quickly mentioned but not enough discussed: in some way, the big and small Al-Idrissi volcanoes seem to play the role of "nucleation" points and to focus strain: is it what you suggest? How to explain the occurrence of such a volcanic activity during the major compressional phase? How do you imagine to infill the syncline axis during folding (l. 300)? Why do you suspect the same age for volcano-clastic deposits and volcanics north of the Alboran Ridge (l. 302)? In Fig. 15b, you suggest subsidence initiation along the big Al-Idrissi volcano: is it linked to the syncline formation during folding? - I have a problem to clearly identify the left–lateral deflections of the hinge axis of the Pliocene folds (l. 235) in Fig. 10: it is not so obvious from your drawing. The differences between this structural sketch and the ones on Fig. 15a and 15b are large. This is important because this pattern is assumed to support the left-lateral transpression in the SAR. - although the left-lateral transtension along the AIF is well evidenced, the southward propagation of the AIF toward the Nekor fault is claimed by the authors but not documented in their study: the role played by this propagation in the change of strain pattern in the area is not a new feature and has been already described in previous studies. For these reasons, I recommend publication after major revision. A rewriting of the manuscript is also necessary to improve the English and to address the many mistakes left behind (sentences without verbs or with two verbs, incorrect grammar, etc...; confusion between terms: compressive or extensive for compressional

and extensional, register for recording, channels for drifts, mass transport complex for MTD with a complex internal structure, northern for western in Figure 14 caption, etc). Figures must be logically presented and all line positions must appear on Figure 3. A Table is also needed at the end in order to summarize the time line and main tectonic and volcanic events with respect to the main structures of the area. Figure 15 should clearly display opposite left-lateral double arrows and mention in Figure 15a the Nekor fault and the SAR.
* * *

---

## Author Comment (AC1) · 13 Nov 2019

*In this paper the authors propose a Plio-Quaternary tectonic evolution of the southern margin of the Alboran Sea, mainly based on the interpretation of multichannel seismic reflection profiles and other relevant stratigraphic information and multibeam data. They identify at least two evolutionary phases for this area: a first ,mostly compressive phase of Tortonian age, ending during the early Quaternary, with a remarkable development of imbricated folds and local occurrence of volcanism and strike-slip structures; (ii) a strike-slip phase with a significant extensional component, which started after 1.8 Ma; within this phase, an important role has been played by the Al-Idrissi fault zone*

*(AIF), which splits the Alboran ridge and, according to the author's interpretation, may represent the present-day plate boundary between Africa and the Alboran domain. As a general comment, the paper should be significantly reduced in length and better organized, especially in the data presentation: in the present form, it is quite difficult to follow the text because it imposes to flip from one figure to another. This because the location maps are distributed in several figures, and for this reason it is difficult to understand the key points presented by the authors. I suggest presenting a single index map in which all the presented profiles are indicated in bold. In the paragraph Data, the authors should present the summary of the acquisition parameters used, and the data processing done (better in a dedicated Table).*

We agree with the referee, and as suggested by the referee 2, we add the missing lines in figure 3. We thank the referee for the suggestion of presenting the dataset in a dedicated table. Yet, we believe that the Materiel and Method section is short enough as it presently is. However, we have tried to enhance the clarity of this section.

*Another point, which should be clarified, is the relationship between the Quaternary subsidence and the strike-slip tectonics, as presented in paragraph 4.2.1. Indeed, it is not clear from the text. Is this part important in the general context of the paper?*

In many strike-slip contexts, vertical motions are associated with faults activity (e.g. the Dead Sea Basin along the Levant fault, Smit et al., 2010). In this paper, we propose that the inception of the transgression of the continental shelf is linked to the propagation of the Al-Idrissi fault. It is important for the general context of this paper because it marks the change of a tectonic regime from the Pliocene to the Quaternary. In a regional context, it is a contribution to date the start of the inception of the present-day Eurasia Africa plate boundary as proposed in Gràcia et al., (2019). We have tried to enhance the clarity of this point.

*In addition, I do not see the importance of the question raised in the paragraph 4.3. It is quite obvious that changes in tectonic styles (and the consequent structural elements*

[Figure]

*produced) are related to changes of direction of stress field.*

We agree with the referee. The title was confusing. In the corrected version, we changed the title of this section to "Evolution of the southeastern limit of the Betico-Rifian tectonic domain." . It is indeed well known that changes in far-field stresses can cause changes in local tectonic regime. It has been proposed for example to explain the change of tectonic regime along the Alboran Ridge during the Pliocene and Quaternary (Comas et al., 1999; Martínez-García et al., 2013). It was partly based on the idea that a change in Nubia-Eurasia-America plate motion occurred after 3.16Ma (Calais et al. 2003). However, more recent plate tectonic models suggest that the direction of convergence is constant through time since 6Ma (DeMets et al. 2015). Two recent publications (Galindo-Zaldivar et al., 2018; Gràcia et al., 2019) propose that the Al-Idrissi fault zone is a recent feature, and corresponds to the present-day Africa-Eurasia plate boundary. Therefore, the questions are how can we explain the change of local tectonic, and can we propose a model explaining the evolution of the Alboran tectonic and the inception of the Al Idrissi fault zone.

*The authors should identify what are the most important structural elements derived from the interpretation, and propose a plausible mechanism, avoiding such a long discussion.*

We agree with the referee. The discussion was indeed a bit long and confuse. Following the remark of the referee 1, we added a table to clarify the timeline of the tectonic events. However, we do not agree with the referee, evidences of rotations of the Alboran Basin are well established (Crespo-Blanc et al., 2016) and it must influence the evolution of the major faults.

*Some of the seismic profiles cross the AIF. Apparently, in the presented data it does not appear as a regional, relevant strike-slip fault accommodating the oblique movement between the two domains.*

We agree with the referee, the AIF appears to produce a very limited lateral displacement (<10km). We argue that it is the consequence of its young age and that the overall regional deformation rates are low (Gràcia et al., 2019; Martínez-García et al., 2017).

*In the earthquake distribution map, the main clusters are located to the south, within the Nekor Basin, and only a few events are located at the Alboran ridge. Have you an interpretation about this?*

Our interpretation is that the main active structure is the Al-Idrissi fault zone. The many strike-slip earthquakes (Mw >6) demonstrate it (Buforn et al., 2004, 2017). This fault zone shifts the NAR and the SAR: the SAR is passively extruded southwestward and the deformation localized on the Nekor and Fez faults to the South of the Rifian Massif. The eastern bloc of the AIF is transported northward causing compression in the Central Alboran Sea (e.g. Estrada, et al. 2018). A compressive foreshock during the last seismic crisis (e.g. Buforn et al. 2018) indicates a local reactivation of compressive structure along the SAR, yet in the immediate vicinity of the AIF, and only in relation to the activation of strike-slip segments of the AIF.

*Possibly we are facing not a single structure, but most probably a sequence of subvertical lineaments distributing the strain, or, in other words, a diffuse transfer zone.*

We agree with the referee. Several fault segments distribute the deformation northward of the studied area in the Adra fault zone and may continue southward affecting the Alboran Ridge and the studied area. Present-day indentation of the central Alboran Sea is likely accommodated through several transfer faults continuing southward (Estrada et al., 2018; Galindo-Zaldivar et al., 2018). However, to the exception of the AIF, these incipient fault zones are not visible at the seafloor in the studied area and their activity is not demonstrated. Gravity anomaly interpreted as incipient faults in Galindo-Zaldivar, et al. 2018 could also correspond to underplated magmatic material following tectonic corridors. Yet, we believe that this discussion is out of the scope of the present paper.
*Most of the presented seismic profiles contain the interpretation superposed. This makes difficult for the reader to follow the interpretation and verify its goodness. For this reason, I suggest to present the uninterpreted and interpreted version of the lines (as an example, the profiles in Fig. 6). Finally, I suggest having the text corrected by a native English speaker. Based on the above, I suggest a moderate revision for this paper.*

We agree with the referee. We will give the uninterpreted version of the seismic lines in supplementary material. We will present a modified figure according to referee's comment.

*Specific points:*

*Line 41: takes control?*

We modified the sentence accordingly: "The Africa-Eurasia NW-SE oblique convergence leads to a tectonic reorganization during the Late Miocene (Comas et al., 1999; Do Couto et al., 2016)."

*Line 52-53:...around 8-7 Ma in the...*

We replaced the text by "7-8 Ma"

*Line 62:Dot after the ) Line 64: use the acronym WAB*

We modified the text according to the referee's comment.

*Line 66: delete "this"*

We modified the text accordingly.

*Line 77:...of thesouthwestern...*

We modified the text accordingly.

*Line 78: We analyse*

We modified the paragraph: "In the present work, we address the structural evolution of the southwestern margin of the Alboran Basin toward the termination of the TASZ through the Plio-Quaternary. We analysis multi-resolution 2D seismic reflection data, TOPAS profiles, and multibeam data. Based on our recent dataset and our seismic stratigraphic interpretation, we observe that the structural subdivision of the Alboran Basin and its margin may reflect a Pleistocene change in tectonic style. We propose a new tectonic model explaining the evolution of the SAR and the Al-Idrissi fault Zone in the southern margin of the Alboran Basin."

*Line 84: write "is", instead of "corresponds to"*

We did not modify this part.

*Line 108 and 111: extensional*

We modified the text accordingly.

*Line 129: has triggered?*

We modified the text and this part of the sentence has been deleted

*Line 133: relative displace-ment? Please explain which plates are involved*

We modified the text accordingly: "GPS kinematics shows a WNW-ESE convergence rate of 4.6mm/yr between Africa and Eurasia plates (Nocquet and Calais, 2004)"

*Line 160: multichannel seismic profiles*

We modified the text accordingly.

*Line 171:...interpretation to perform the...*

We modified the text accordingly.

*Line 178: why you use this acoustic velocity? Have you performed velocity analyses on the data? This should be clarified*

We did not perform velocity analysis, we used the velocity analysis for the ODP well 976 from Soto et al. 2012. It is now clarified in the paper.

*Line 183: sedimentary sequence*

We modified the text accordingly.

*Line 192: evidence*

We modified the text accordingly.

*Line 198:...to a 4-8 km wideconic...*

We modified the text accordingly.

*Line 199: This sentence is unclear...rewrite*

This sentence has been modified to be more concise and clearer.

*Line 212: unconformably*

We modified the text accordingly.

*Lines219-220: conversion in depth based on what velocity? (see comment above)*

We did not perform velocity analysis, we used the velocity analysis for the ODP well 976 from Soto et al. 2012. It is now clarified in the paper.

*Line 225:this sentence is unclear... Line 241: this sentence is unclear...Line 247: ???*

We reworked the paragraph to be more concise and to enhance its clarity.

*Line254:...fault zone composed by...*

We modified the text accordingly.

*Line 290: why sinistral shear? Explain*

We moved this sentence and reformulate.

*Line 292:..Quaternary at..*

The text has been modified to : "The aggradation of contourites at the foot of the SAR indicates a relative quiescence of the folding during the Quaternary after 2.6 Ma (Juan et al., 2016)"

*Line 295: equivalent volcano? This is unclear...re-phrase*

We rephrase: "The lateral continuity of the highly reflective facies from west to east suggests that the Small Al-Idrissi and the Big Al-Idrissi volcanoes are continuous structures which are offset by local extensional faults during the Pleistocene (Fig. 8 and 10)."

*Line 304:...could be the product of MCS...(please explain the acronym and the meaning of this).*

We corrected the typo.

*Line 309: paleo-ria???*

From Wikipedia: A submergent coastal landform, often known as a drowned river valley.Please note that the expression 'paleo-ria' is from Romagny et al., (2014).

*Line 325: implies*

We reformulate and the comment is not valid anymore.

*Line 336: Evidence of...*

We reformulate and the comment is not valid anymore.

*Caption of Fig. 2:...modified from and from...???*

We modified the caption accordingly.

*Caption of Fig. 3: indicate NB,BB,..*

We modified the caption accordingly.
*Fig 6: please present uninterpreted and interpreted profiles, at a larger scale!*

We will present the uninterpreted profiles in the supplementary materials

*Caption of Fig. 8: there is a repletion of sentence (the seismic line shows.)*

We modified the caption accordingly.

*Fig. 9: I donot see the location of this figure*

The position of the line was given in the figure 3 and 10. However we modified the figure 3 for clarity.

**Bibliography**

Buforn, E., Bezzeghoud, M., Udìas, A. and Pro, C.: Seismic Sources on the Iberia-African Plate Boundary and their Tectonic Implications, Pure and Applied Geophysics, 161(3), 623–646, doi:10.1007/s00024-003-2466-1, 2004.

Buforn, E., Pro, C., Sanz de Galdeano, C., Cantavella, J. V., Cesca, S., Caldeira, B., Udías, A. and Mattesini, M.: The 2016 south Alboran earthquake (M w = 6.4): A reactivation of the Ibero-Maghrebian region?, Tectonophysics, 712–713, 704–715, doi:10.1016/j.tecto.2017.06.033, 2017.

Comas, M. C., Platt, J. P., Soto, J. I. and Watts, A. B.: The origin and Tectonic History of the Alboran Basin: Insights from Leg 161 Results, Proceedings of the Ocean Drilling Program Scientific Results, 161, 555–580, 1999.

Crespo-Blanc, A., Comas, M. and Balanyá, J. C.: Clues for a Tortonian reconstruction of the Gibraltar Arc: Structural pattern, deformation diachronism and block rotations, Tectonophysics, 683, 308–324, doi:10.1016/j.tecto.2016.05.045, 2016.

Do Couto, D., Gorini, C., Jolivet, L., Lebret, N., Augier, R., Gumiaux, C., d'Acremont, E., Ammar, A., Jabour, H. and Auxietre, J.-L.: Tectonic and stratigraphic evolution of the Western Alboran Sea Basin in the last 25 Myrs, Tectonophysics, 677–678, 280–311, doi:10.1016/j.tecto.2016.03.020, 2016.

Estrada, F., Galindo‐Zaldívar, J., Vázquez, Gemma, E., D'Acremont, E., Belén, B. and Gorini, C.: Tectonic indentation in the central Alboran Sea (westernmost Mediterranean), Terra Nova, 30(1), 24–33, doi:10.1111/ter.12304, 2018.

Galindo‐Zaldivar, J., Ercilla, G., Estrada, F., Catalán, M., d'Acremont, E., Azzouz, O., Casas, D., Chourak, M., Vazquez, J. T., Chalouan, A., Galdeano, C. S. de, Benmakhlouf, M., Gorini, C., Alonso, B., Palomino, D., Rengel, J. A. and Gil, A. J.: Imaging the Growth of Recent Faults: The Case of 2016–2017 Seismic Sequence Sea Bottom Deformation in the Alboran Sea (Western Mediterranean), Tectonics, 0(0),
doi:10.1029/2017TC004941, 2018.

Gràcia, E., Grevemeyer, I., Bartolomé, R., Perea, H., Martínez-Loriente, S., Peña, L. G. de la, Villaseñor, A., Klinger, Y., Iacono, C. L., Diez, S., Calahorrano, A., Camafort, M., Costa, S., d'Acremont, E., Rabaute, A. and Ranero, C. R.: Earthquake crisis unveils the growth of an incipient continental fault system, Nat Commun, 10(1), 1–12, doi:10.1038/s41467-019-11064-5, 2019.

Juan, C., Ercilla, G., Javier Hernández-Molina, F., Estrada, F., Alonso, B., Casas, D., García, M., Farran, M., Llave, E., Palomino, D., Vázquez, J.-T., Medialdea, T., Gorini, C., D'Acremont, E., El Moumni, B. and Ammar, A.: Seismic evidence of current-controlled sedimentation in the Alboran Sea during the Pliocene and Quaternary: Palaeoceanographic implications, Marine Geology, doi:10.1016/j.margeo.2016.01.006, 2016.

Martínez-García, P., Comas, M., Soto, J. I., Lonergan, L. and Watts, A. B.: Strike-slip tectonics and basin inversion in the Western Mediterranean: the Post-Messinian evolution of the Alboran Sea, Basin Research, 25(4), 361–387, doi:10.1111/bre.12005, 2013.

Martínez-García, P., Comas, M., Lonergan, L. and Watts, A. B.: From extension to shortening: tectonic inversion distributed in time and space in the Alboran Sea, Western Mediterranean: Tectonic inversion in the Alboran Sea, Tectonics, doi:10.1002/2017TC004489, 2017.

Nocquet, J.-M. and Calais, E.: Geodetic Measurements of Crustal Deformation in the Western Mediterranean and Europe, Pure and Applied Geophysics, 161(3), 661–681, doi:10.1007/s00024-003-2468-z, 2004.

Smit, J., Brun, J.-P., Cloetingh, S. and Ben-Avraham, Z.: The rift-like structure and asymmetry of the Dead Sea Fault, Earth and Planetary Science Letters, 290(1), 74–82, doi:10.1016/j.epsl.2009.11.060, 2010.

---

## Author Comment (AC2) · 13 Nov 2019

*Jacques Déverchère (Referee)*

*jacdev@univ-brest.fr*

*This paper deals with the recent tectonic evolution of part of the southern margin of the Alboran Basin (and not the whole southern margin as wrongly stated in the title). Actually, the authors focus on the offshore southernmost part of the Trans-Alboran Shear Zone (TASZ) representing a broad area of deformation which is not as well*

*documented as other areas in the Alboran domain. The interest of the work is to further document this area with new, high-quality seismic data of high to very high resolution and to better assess and/or discuss the reasons for the fast stress changes that occurred since Pliocene. As a whole, this contribution appears stimulating and rather convincing and is worth to be published.*

*However, several limitations appear in the way the authors reports previous studies and discuss their interpretations; furthermore, the bad organization of the fi̧gures and the poorly written English make the paper quite diffi̧cult to read.*

*Actually, I have identifi̧ed an important weakness: an important point supported by the authors is to suggest that besides the well-assessed processes of indentation in the central Alboran domain and extrusion of the Rif, the major change of tectonic style in the study area is related to a clockwise rotation of the Alboran tectonic domain instead of a change in the Eurasia/Africa plate convergence vector. The reality is that (1) uncertainties on kinematic data (DeMets at al., 2015) prevent from assessing any signifi̧cant change in the obliquity angle or in slip direction in the study area since the Messinian, although Nubia-Eurasia angular velocities estimated from geodetic and geologic observations appear to differ signifi̧cantly. (2) the block rotation model proposed by Meghraoui and Pondrelli (2013) is a large-scale model ("restraining bend") based on the assumption of a right-lateral deforming zone with large fault systems in the offshore domain: in my opinion, the fault geometry, sense of motion and continuity are far from being clearly assessed, preventing from concluding fi̧rmly that a block rotation model (either bookshelf or pinned) is responsible for the change in tectonic style. It results from (1) and (2) that the choice made by the authors is quite questionable, in my opinion. Other parameters such as, for instance, changes of body forces during crustal thickening of the South Alboran Ridge or even further south, or transpressive fold propagation (as suggested, lines 403-404), or also effects of propagation of the Al-Idrissi fault on strain distribution, should be included in the discussion on this challenging question.*

We agree with the reviewer and thank him for his constructive comment. He high-lights issues that might represent critical weakness in our paper. From the results of DeMets et al., (2015), the Africa-Eurasia convergence azimuth 95% confidence inter-val is approximatively around +/- 7° at 2Ma; the error on the velocities are inferior to 1mm/a. More important is that the trends of the velocity and the azimuth are stable since 6Ma. Estimates of local changes in stress direction since 5.33Ma are around 12-15° (based on structural data, Martínez-García et al., 2013). This is the same order than the amount of rotation from pinned block model (Meghraoui and Pondrelli, 2013) and in the same order of magnitude of strain rotation in Palano et al. 2013, but also in the same order of rotation in the Betic-Rif orogenic arc (Crespo-Blanc et al., 2016). A well-described 20° difference exists between the direction of Africa/Eurasia conver-gence given by GPS and kinematic models (e.g,. Bougrine et al., 2019). Calais et al., (2003) propose that a change in Africa/Eurasia kinematic after 3.16 Ma may explain it. As discussed in DeMets et al. 2015, a recent (maybe younger than a few hundred thousand years) changes of kinematic can explain the discrepancy between long-term kinematic model and GPS data. Yet, it implies rapid changes in moments of forces that could be impossible.

We would like the referee to consider the following reasoning: any change of the Africa-Eurasia rates and directions of convergence that is recorded in the tectonic structures of the Alboran Basin must be recorded in similar structures within the Mediterranean Basin. Considering that Nubia is a rigid block, we argue that any changes (i.e,. period <5My) of Eurasia/Nubia convergence direction or important change in angular veloc-ities must lead to a tectonic reorganization at the scale of the Mediterranean. Sedi-mentary basins must record such a tectonic reorganization. If indeed, some pieces of evidence of rejuvenation of vertical motion along East Mediterranean basins exist around the Mid-Pleistocene Revolution in the Eastern Mediterranean (Gawthorpe et al., 2018; de Gelder et al., 2019; Schattner, 2010), it is difficult to relate those changes to a change of Africa/Eurasia convergence. It is out of the scope of this paper to dis-cuss if such reorganization exists at the scale of the whole Mediterranean Basin, and

if it exists, if it linked to plate tectonic, to change of slab behavior in the mantle or to climate change.

Some published and unpublished 3D numerical models (Le Pourhiet et al., 2014) demonstrate that, in a transpressive setting, main shear zones rotate until new better-oriented shear zones start to nucleate, as proposed from analytical models of blocks rotations (Nur et al., 1986; Ron et al., 2001). Following the Occam's razor principle, we assume that the tectonic reorganization in the Alboran basin occurring during the Pleistocene is linked to local tectonic evolution and rotation of the Alboran tectonic domain, rather than a change of kinematic. A similar model is also proposed from onshore field studies (Crespo-Blanc et al., 2016).

On the second point, we agree that the geometry of the model of Meghraoui and Pondrelli (2013) is far from being evident. Yet, their estimations are close to estimates of block rotation from other studies (Crespo-Blanc et al., 2016; Platt et al., 2003). The geometry of the right lateral fault is not to be demonstrated. A right-lateral strike-slip fault zone exists onshore in the Beti,c accommodating the extrusion of the Betic-Rif tectonic domain (Galindo-Zaldivar et al., 2015; Gonzalez-Castillo et al., 2015). The continuity of this right lateral fault zone toward the East is complex because it seems to form a triple junction with the Carboneras and Palomares fault zones. Its left lateral behavior is evidenced by GPS data (Galindo-Zaldivar et al., 2015). A right-lateral strike-slip structure occurs offshore further South, from the Adra fault zone to the Yusuf Fault and then to the Algerian Margin (e.g,. Estrada et al., 2018; Martínez-García et al., 2011). The northern segment of this structure is younger than 1.1Ma (Perea et al., 2018). On a regional scale, it draws a poorly continuous shear zone from the External Units in the Betic Belt to the Algerian Margin.

Furthermore, we agree with the referee, changes of body forces may have occurred. However, we must then examine the causes of such changes. A first change could be the Messinian Salinity Crisis. An isostatic response to the 1500m change of water level must have some consequences on the local tectonic. It is not clear how changes

of vertical stress overtime in the lithosphere modify the local compressive tectonic and to our knowledge, this is not known. Another cause of change of body force can be upper-lithospheric mantle and/or lower crustal delamination associated to sinking lithospheric material (e.g. Calvert et al., 2000; Levander et al., 2014; Petit et al., 2015) and lithosphere tearing at STEP fault (Hidas et al., 2019). Slab rollback as the main driving force is likely to have ended or to be a considerably weak force during the Pliocene as the age of youngest evidence of accretion in the Rif is Upper-Miocene (Capella et al., 2016). Upper mantle and lower crust delamination can occur below the Betico-Rifian tectonic domain (Petit et al., 2015). It can influence the velocity field as demonstrated for thermomechanical modeling (Le Pourhiet et al., 2006), thought viscous coupling (Perouse et al., 2010; Petit et al., 2015). Regional-scale changes of buoyancy force through changes of coupling or delamination can indeed happen from the Pliocene, yet it would have produced a change of vertical motion pattern along the Rifian arc.

To our opinion, uplift and radial extension in the central Rif can evidence such a Pliocene change of body forces. It is possible that this mechanism contributes to the regional stress-field and to WSW-ENE extrusion of the Rif. In addition to the extrusion, body forces can contribute to the localization of the Al-Idrissi fault along an area of a strong gradient of crustal thickness. By decoupling the deformation across the Alboran Ridge, the inception of the Al-Idrissi Fault during the Quaternary zone can trigger a latter change of strain portioning and pure compression along the SAR and the NAR. It might be possible to demonstrate it by thin shell modeling. This is part of an ongoing investigation using the method described in Pérouse et al., (2012), and a complete discussion of this last mechanism is out the scope of the present paper.

 *I have other comments below: - The introduction part is full of unclear statements (i.e., lines 79-80, 81-82, 129-130, etc. . .), so that facts are often confused and mixed with interpretations. I recommend carefully check the reported assessments in order to clarify between what is established and what is an hypothesis. –*

We agree with the referee. We modified the introduction to do a clear distinction between hypothesis and data.

*The term "inversion" is systematically used throughout the paper; however, it appears that not only inversion is actually occurring, but merely reactivation, or tectonic reorganization. I urge the authors to distinguish a true tectonic inversion of structures (or of the margin) from other types of tectonic changes (for instance, from a strike-slip to an extensional strain change).*

We agree with the referee. Indeed, we used the word inversion too often. We rephrase to avoid such terminology when needed. We would like to clarify the following point. The Alboran Basin is a fore-arc basin during the Miocene times (Duggen et al., 2008; Peña et al., 2018). The studied area is the southern termination of the WAB, which corresponds to the STEP fault. The exact geometry of this STEP when the slab-roll back slows down is yet not well known. This topic is the subject of a new paper already submitted to Tectonics. The tectonic reorganization described in this paper is, however, younger than the tectonic inversion that starts during the Miocene around 8Ma.

*I also urge the authors to better highlight the new features and facts they bring compared to previous studies instead of speculating too much on a possible cause of strain re-organization.*

We agree with the referee. According to referee 1 and to the referee 2, in the revised version, we propose a synthesis of the tectonic events in the Betico-Rifian arc and Alboran basin. This allows a better comparison with the recent papers published on the neotectonics of the Alboran Basin.

*For instance, the clear change of strain pattern (both in tectonic style but also in strain expression) shown on Figure 9 between Pliocene and Quaternary is an excellent example of strain superimposition at short time scale on a same geological structure and deserves for instance a comparison with the well-expressed NW-SE set of diffuse, secondary faults identified between the Carboneras fault and the north Alboran Ridge further north (Perea et al.,Marine Geology, 399, 23-33, 2018).*

The referee is right. The dataset in Perea et al., (2018) allows proposing a timing of propagation of secondary NW-SE dextral strike slip faults of the Yusuf fault system, which fits well with our results. These authors propose that early Pliocene subsidence occurred in the Djibouti plateau. After a period of tectonic quiescence, strike-slip faulting starts during the Quaternary. The propagation of NW-SE strike-slip faults occurs around 1.1 Ma. It fits with our proposition that the inception of the deformation along the AIF is younger than 1.8 Ma and should start around with the rapid transgression of the shelf in the Nekor basin after 1.12 Ma. In our interpretation, fault localization along present-day active segments of AIF occurs after 0.8Ma (Lafosse et al. 2017). We modified the discussion according to the referee's comment.

*- The role of volcanism in the tectonic evolution is only quickly mentioned but not enough discussed: in some way, the big and small Al-Idrissi volcanoes seem to play the role of "nucleation" points and to focus strain: is it what you suggest? How to explain the occurrence of such a volcanic activity during the major compressional phase?*

The referee is right, yet following the comment of referee one, the length of our discussion is already too important. We suggest that poorly dated volcanism occurred in the Early-Pliocene. The volcanism could occur as a response to the crustal thinning (Duggen et al., 2008) or it could also be a response to changes of body forces after the Messinian Salinity Crisis (Sternai et al., 2017). Gravity anomaly interpreted as incipient faults in Galindo-Zaldivar, et al. (2018) could also correspond to underplated magmatic material following pre-existing tectonic corridors. Yet, we believe that this discussion is out of the scope of the present paper. We can hypothesis that local volcanism occurs in response to mantle delamination and to an early-inception of strike-slip faults. We can only conjecture that the volcanism localizes along tectonic features as in another context (e.g. the North Sea graben, Quirie et al., 2018).

*How do you imagine to infił the syncline axis during folding (l. 300)?*

We interpret that volcanic activity and the subsequent deposition of volcanoclastic sediments are guided by the geometry of the fold NE-SW trend.

*Why do you suspect the same age for volcano-clastic deposits and volcanics north of the Alboran Ridge (l. 302).*

We suspect the same ages because the volcanism North of the Alboran Ridge, which seems to have similar chemistry than the samples studied in the Francesc Pages high, is dated from early Pliocene (Duggen et al., 2008). It corresponds to the stratigraphic age constraints that we have for the volcanism of the studied area. However, we agree that is speculative.

*In Fig. 15b, you suggest subsidence initiation along the big Al-Idrissi volcano: is it linked to the syncline formation during folding?*

We thank the referee for his remark. From our interpretation, the subsidence is linked to normal faulting. Before 1.8Ma, the basinward motion of the shoreline and the normal regressive geometry of the shelf argue for progradation driven by sediment supply. Sedimentation rates outpace the rates of relative sea-level rise (positive accommodation) at the coastline (Catuneanu et al., 2011). It is likely that the overall regressive trend is linked to positive accommodation space during syncline formation. However, the later Pleistocene transgression is clearly linked to the normal faulting. We modified the text according to the reviewer comments.

*I have a problem to clearly identify the left–lateral deflections of the hinge axis of the Pliocene folds (l. 235) in Fig. 10: it is not so obvious from your drawing. The differences between this structural sketch and the ones on Fig. 15a and 15b are large. This is important because this pattern is assumed to support the left-lateral transpression in the SAR.*

We agree with the reviewer comment, interpretations of left lateral deflection of the hinge axis are difficult. However, we can simply show that Pliocene folds are oblique to the general N065° of the Alboran Ridge. To convince the referee, we can also

compute from our drawing the azimuth of the hinge axis folds at each vertex (Figure 1 below). The resulting figure illustrates the change of orientation of the hinge axis. From analogue model experiments, oblique fold patterns occurred in transpressive/strike-slip shear zones (Richard, 1991). The amount of rotation evolves through time if a function of the mode of transpression (pure/simple shear, Fossen et al., 1994), as demonstrated elsewhere (Tadayon et al., 2018). In figure 15, we draw the simplified fold axis to help the reader.

*Although the left-lateral transtension along the AIF is well evidenced, the southward propagation of the AIF toward the Nekor fault is claimed by the authors but not documented in their study: the role played by this propagation in the change of strain pattern in the area is not a new feature and has been already described in previous studies. For these reasons, I recommend publication after major revision.*

We agree with the referee, and modify the text accordingly, by adding a table of the tectonic event at a regional scale. However, there is two recent papers demonstrating that the Al-Idrissi fault is recent (i.e. Pleistocene) feature (Galindo-Zaldivar et al., 2018; Gràcia et al., 2019) and one paper proposing a model explaining the strain pattern (Spakman et al., 2018). None of those papers proposes new pieces of evidence showing how the Al-Idrissi fault is different from the Trans Alboran Shear Zone in the SAR region. The Miocene to Quaternary tectonics in the NAR and Yusuf fault is described in Martínez-García et al., (2013) and (2017). We propose an age of the Bokkoya fault in Lafosse et al. 2017. To our knowledge, this paper is the first study integrating the data along the AIF and the SAR.

*A rewriting of the manuscript is also necessary to improve the English and to address the many mistakes left behind (sentences without verbs or with two verbs, incorrect gram-mar, etc. . .; confusion between terms: compressive or extensive for compressional and extensional, register for recording, channels for drifts, mass transport complex for MTD with a complex internal structure, northern for western in Figure 14 caption,. etc).*

We modified the text according to the referee's comments.

*Figures must be logically presented and all line positions must appear on Figure 3.*

We modified the figure accordingly.

*A Table is also needed at the end in order to summarize the time line and main tectonic and volcanic events with respect to the main structures of the area.*

We added the table according to the referee comment.

*Figure 15 should clearly display opposite left-lateral double arrows and mention in Figure 15a the Nekor fault and the SAR.*

We modified the figure accordingly.

[Figure]

```
./azimut.pdf
```

**Fig. 1.** : Azimuths of the hinge axis of the folds along the SAR area. Azimuths tend toward the E-W direction at the center of the folds and toward a N065 direction toward the tips of the folds, therefore demonstrating the left lateral deflection. Computations of the strike of the folds hinge axis are done using MatLab.

---

## Author Response (AR1)

**Author's response**

The present revised version of the paper entitled "Plio-Quaternary tectonic evolution of the southern margin of the Alboran Basin (Western Mediterranean)" is an attempt to correct the flaws pointed out by the referee 1 (Dr. Lododo), and the referee 2 (Prof Déverchère), and based on the already published open discussion Here, we would like to thank both referees for their valuable comments.

According to both referees, we made an effort to enhance the phrasing and the clarity of the paper. However, its length does not change, since referee 1 suggested to reduce its size, and the referee 2 suggested to develop some sections and to add a table. In its present shape, the paper is approximatively 8750 words long (including the figure captions, excluding the title and author's sections, the abstract, and the acknowledgments). It falls within the standard lengths of scientific publications.

Some figures have been reworked. According to referee 1's remarks, we now present the position of all the seismic lines in the figure 3. We still present the positions of seismic lines in the following figures when we think it will help the reader. According to referee 2, we add a table 1, which presents tectonic events happening in the Rif and in the Alboran basin between the Pliocene and the present-day. We also modify the figure 15 according to the reviewer remarks.

The following sections of this answer present a detail of the changes made, first, section by section of the original manuscript, then it presents detailed changes according to the referee's remarks. The end of this answer presents the comparison between the original and the revised manuscript.

As required by the Solid Earth editor, we add the compared versions at the end of the present answer.

**1. Changes sections by sections**

**Introduction and geological setting**

The section has been modified according to referees 1 and 2 remarks. We strongly reorganized the paragraphs to comply with the referee 2 remarks.

**Materiel and Methods**

Material and methods have been slightly modified for clarity. According to the referee 1's comment we better present the velocity analyses from Soto et al., (2012).

**Results**

The Results section has been slightly modified. Grammatical and orthographic mistakes have been corrected. Some sentences have been modified to enhanced the clarity of the results.

**Discussion**

30 We strongly modified this section according to the referee 1 and 2 remarks. We reorganised the paragraphs to enhance the clarity of the discussion. We add some discussions according to the referee 2 remarks, as discussed in the previous answer to referee. We now emphasize the evolution of strain partitioning from the Pliocene to the present-day rather than the discussion about the subjacent mechanisms behind this change. We modified the title of the last sub-section to comply with referee 1 remarks. A table showing the succession of the tectonic events at regional scale has been added to support the discussion and

35 to enhance its clarity.

**2.  Detailed changed according to the referee 1 remarks.**

*Specific points:*

*Line 41: takes control?*
We modified the sentence accordingly: "The formation of the Alboran Basin has been linked to back-arc extension and
40 lithospheric tearing during early Miocene (e.g., Jolivet et al., 2009, 2008)." L39-40.

*Line 52-53:...around 8-7 Ma in the...*
We replaced the text by "8 Ma" L46

*Line 62:Dot after the ) Line 64: use the acronym WAB*
We modified the text according to the referee's comment.

45 *Line 66: delete "this"*
We modified the text accordingly.

*Line 77:...of thesouthwestern...*
We modified the text accordingly.

*Line 78: We analyse*
50 We modified the paragraph: "In the present work, we address the structural evolution of the southwestern margin of the Alboran Basin toward the termination of the TASZ through the Plio-Quaternary. We analysis multi-resolution 2D seismic reflection data, TOPAS profiles, and multibeam data. Based on our recent dataset and our seismic stratigraphic interpretation, we observe that the structural subdivision of the Alboran Basin and its margin may reflect a Pleistocene change in tectonic style. We propose a new tectonic model explaining the evolution of the SAR and the Al-Idrissi fault Zone in the southern
55 margin of the Alboran Basin."

*Line 84: write "is", instead of "corresponds to"*
We did not modify this part.

*Line 108 and 111: extensional*
We modified the text accordingly. As required by the referee 2, we corrected the misused words extensional/extensive through
60 the text.

*Line 129: has triggered?*
We modified the text and this part of the sentence has been deleted.

*Line 133: relative displace-ment? Please explain which plates are involved*

We modified the text accordingly: "GPS kinematics shows a WNW-ESE convergence rate of 4.6mm/yr between Africa and Eurasia plates (Nocquet and Calais, 2004)" L139-140.

*Line 160: multichannel seismic profiles*

We did not modified the text.

*Line 171:...interpretation to perform the...*

We modified the text accordingly. L173.

*Line 178: why you use this acoustic ve-locity? Have you performed velocity analyses on the data? This should be clarified*

We did not perform velocity analysis, we used the velocity analysis for the ODP well 976 from Soto et al. 2012. It is now clarified in the paper.

*Line 183: sedimentary sequence*

We modified the text accordingly. L185.

*Line 192: evidence*

We modified the text to: Pl1, Pl2 and Pl3 units are pinching toward the structural highs and evidence aggrading wedges geometries." L188.

*Line 198:...to a 4-8 km wideconic...*

We modified the text accordingly.

*Line 199: This sentence is unclear...rewrite*

This sentence has been modified to be more concise and clearer. "The Big Al-Idrissi Volcano corresponds to a conic structure located to the North of the Ras Tarf ( Fig. 8) that has been interpreted as an N-S volcanic ridge in Bourgois et al. (1992). The top of this seismic body merges with the M-Reflector (Fig. 8)." L202.

*Line 212: unconformably*

We modified the text accordingly.

*Lines219-220: conversion in depth based on what velocity? (see comment above)*

We did not perform velocity analysis, we used the velocity analysis for the ODP well 976 from Soto et al. 2012. It is now clarified in the paper.

*Line 225:this sentence is unclear...*

We modified the sentence to: "Folds and faults along the Alboran Ridge demonstrate a Pliocene compressive phase. On the Moroccan shelf, the stratigraphic pattern indicates a regressive trend. The second phase is younger and corresponds to the developing activity of strike-slip and normal faults, which control the local transgressions of the Moroccan shelf." L228.

*Line 241: this sentence is unclear...Line 247: ???*

We reworked the paragraph to be more concise and to enhance its clarity: "The intra-Pliocene unconformities, the tilting of the Pliocene units, and the aggradation of Quaternary contouritic deposits on top of the sedimentary sequence indicate a compressive deformation ending around the early Quaternary (Fig. 6 and 9). Within the Pliocene sequence, the folding appears to be progressive and diachronic from East to West. At the foot of the Francesc Pagès Bank, P1 reflectors are unconformably lying on the P0 reflector (Fig. 7a). At the foot of the Ramon Margalef High, Pliocene reflectors older than P1 show a more even geometry with constant thickness, where P0 is a conformable surface (Fig. 7b)." L245-250

*Line254:...fault zone composed by...*

We modified the text accordingly.

*Line 290: why sinistral shear? Explain*

We moved this sentence and reformulate. Also following referee 2's remarks, the following sentences are added in the result section: "Azimuths of the hinge axis tend toward the E-W direction at the center of the folds and toward an N065° direction toward the tips of the folds, therefore demonstrating left lateral deflections of their hinge axis and overall sigmoidal shape (Fig. 6 and 10). The western part of the SAR corresponding to the Xauen Bank is deflected toward the E-W direction (Fig. 6). " L238-L241.

A comparison to other papers (e.g. Koyi et al., 2016; Tadayon et al., 2018) allow proposing a sense of shear in the discussion section.

*Line 292:..Quaternary at..*

The text has been modified to : "The aggradation of contourites at the foot of the SAR indicates a relative quiescence of the folding during the Quaternary after 2.6 Ma (Juan et al., 2016)" L290-291.

*Line 295: equivalent volcano? This is unclear...re-phrase*

We rephrase: "The lateral continuity of the highly reflective facies from west to east suggests that the Small Al-Idrissi and the Big Al-Idrissi volcanoes are continuous structures which are offset by local extensional faults during the Pleistocene (Fig. 8 and 10)."

*Line 304:...could be the product of MCS...(please explain the acronym and the meaning of this).*

We corrected the typo.

*Line 309: paleo-ria???*

From Wikipedia: A submergent coastal landform, often known as a drowned river valley. Please note that the expression 'paleo-ria' is from Romagny et al., (2014).

*Line 325: implies*

We reformulate and the comment is not valid anymore.

*Line 336: Evidence of...*

We reformulate and the comment is not valid anymore.

*Caption of Fig. 2:...modified from and from...???*

We modified the caption accordingly.

*Caption of Fig. 3: indicate NB,BB,..*

We modified the caption accordingly.

*Fig 6: please present uninterpreted and interpreted profiles, at a larger scale!*

We will present the uninterpreted profiles in the supplementary materials

*Caption of Fig. 8: there is a repletion of sentence (the seismic line shows.)*

We modified the caption accordingly.

*Fig. 9: I donot see the location of this figure*

The position of the line was given in the figure 3 and 10. However, we modified the figure 3 for clarity.

**3. Detailed changed according to the referee 2 remarks.**

*The introduction part is full of unclear statements (i.e., lines 79-80, 81-82, 129-130, etc. . .), so that facts are often confused and mixed with interpretations. I recommend to carefully check the reported assessments in order to clarify between what is established and what is an hypothesis.*

The introduction has been reworked according to the reviewer remarks.

*A rewriting of the manuscript is also necessary to improve the English and to address the many mistakes left behind (sentences without verbs or with two verbs, incorrect gram-mar, etc. . .; confusion between terms: compressive or extensive for compressional and extensional, register for recording, channels for drifts, mass transport complex for MTD with a complex internal structure, northern for western in Figure 14 caption, etc).*

We modified the text according to the referee's comments. However, in the text and in the figures, the word drift is used to described the depositional part of coutouritic sedimentary featues as decribed in the literature (e.g., Hernández-Molina et al., 2008).

*Figures must be logically presented and all line positions must appear on Figure 3.*

We modified the figure accordingly.

*Table is also needed at the end in order to summarize the time line and main tectonic and volcanic events with respect to the main structures of the area.*

We added the table according to the referee comment.

*Figure 15 should clearly display opposite left-lateral double arrows and mention in Figure 15a the Nekor fault and the SAR.*

We modified the figure 15 accordingly.

**4.  Compared versions of the manuscript.**

[revised manuscript text omitted]

---

## Referee Report (RR1)

[referee-annotated manuscript omitted]

---

## Editor Decision (ED1)

[revised manuscript text omitted]

Big Al - Idrissi volcano

−600m

Francesc pages Bank

−500m
−400m
−300m
−200m
−100m

N145° fault zone

fig. b

fig. 9

Al - Idrissi fault splay

Bokkoya fault

Small Al - Idrissi volcano

Fig. 8

Ras Tarf

Boudinar fault

0 1 2 3 4 5 km

35.65
35.60
35.55
35.50
35.45
35.40
35.35

−3.90   −3.85   −3.80   −3.75   −3.70   −3.65   −3.60

b) SW   **N145° fault zone**   NE

100 mstwt

2km

**Figure 11: Active structures around the roughly NNE-SSW AIF and adjacent submarine highs. The AIF bends to the North, where it follows the trends of the NAR. High values of curvatures in the Francesc Pagès Bank and the Northeast corner of the map underline the linear features at the seafloor, which corresponds to the truncated Miocene-Pliocene layers. Extreme positive values in red represent concave topography at the seafloor; extreme negative values in blue represent convex topography. a) Profile curvature map textured above the shaded bathymetry; dashed purple lines, fault tracks at the seafloor; dashed black lines, positions of the seismic line in (b) and in figures 8, 9, and 13. b) TOPAS profile showing active N145° normal faults. Red lines, active faults; red arrows, positions of the fault traces in (a).**

880

[Figure]

Figure 12: SPARKER seismic line showing the transgression of marine sediment (in green) over the prograding shelf of the Nekor Basin (in pink). Oldest depositional units (Pliocene) are colored in blue and the acoustic basement in grey. The Maximum Regressive Surface (MRS) is in red.

[Figure]

890

**Figure 13: Multichannel seismic lines across the restraining bend of the Al-Idrissi fault zone showing lateral evolution of the tectonic structures in North Alboran Ridge and in the restraining bend. a) The Al-Idrissi fault zone is a positive flower structure following the front of the Alboran Ridge. b) The Al-Idrissi fault zone is a positive flower structure distinct from the Pliocene thrusts and folds.**

[Figure]

895

**Figure 14: Active structures affecting the northern flank of Francesc Pagès and Ramon Margalef highs. a) plan curvature map overlying the shaded bathymetry; red arrows pockmarks on the seafloor; dashed black lines, seismic lines in the figures (b) and (c); dashed red lines, positions of the fault tracks. b) SPARKER seismic reflection line showing the northward continuity of N145° fault (red line). c) TOPAS seismic line showing the subsurface of the seafloor. Red arrows, positions of the faults drawn in a).**

900

[Figure]

(a)

vertical axis rotation

SLAB

AIF

SAR

CR

JF

NF

Yusuf fault

SAR

future
Al-Idrissi
fault zone

Nekor Fault

Early Pliocene

(b)

vertical axis rotation

SLAB

AIF

CR

JF

NF

Transtensive initiation

Early Pleistocene

(c)

vertical axis rotation

SLAB

AIF

AIF

CR

JF

NF

Present - Day Nekor Basin

Present-Day

[Figure]

**Figure 15: Palinspastic maps of the SAR and the Rif from 5 Ma** to the present-day are using 14 ° clockwise rotation of the Alboran tectonic domain from a) to c). **Dashed blue line, approximate coastline; continuous blue line, present-day coastline; Dark yellow, Miocene-Pliocene onshore basins; light yellow, Pliocene and Quaternary onshore basins; grey patch, position of the slab remaining approximatively constant below the Alboran Basin during the Plio-Quaternary; left bottom corner of the maps, simplified drawing figure the area between the SAR, the Nekor fault and the Yusuf fault. Thick grey arrows in (c) indicate the direction and relative amount of extrusion in the central Rif considering a fixed Eurasia. The shortening is accommodated through compressive structures in (a). The initiation of subsidence along the Big Al-Idrissi Volcano and the Moroccan shelf corresponds to (b), and the present–day partitioning of the deformation corresponds to (c). CR, central Rif, JF, Jebha Fault; NF, Nekor Fault; AIF, Al-Idrissi Fault zone.**

---

## Author Response (AR2)

**Cover letter**

Dear Editor,

The present revised version of the paper entitled "Plio-Quaternary tectonic evolution of the southern margin of the Alboran Basin (Western Mediterranean)" is an attempt to correct the flaws pointed out by the referee 3 (Guillermo Booth-Rea) and the R1. As pointed out in the editor review, background information was lacunose in some parts of the introduction, and we did not adequately discussing any alternative model for the tectonic evolution of the region. As indicated by the R1 and R3, some typing errors were present and have been corrected. In the text and in the figures 10, 11 and 13, we replaced the words *compressive bend*, by *restraining bend.*

In the introduction, we precise the ages of the inversion and the extensional structures in the West Alboran Basin. We also introduce results from R3 papers. In the discussion, we now clearly discuss the effect of lithospheric sinking and necking of the structural pattern.

In his review, R3 points to the possible influence of lithospheric tear and delamination processes. We agree with the referee that deep mantellic processes are important and we clarify our argumentation. We made small modifications in the Section 4.1 to better discuss the role of crustal thinning occurring north of the Alboran basin during the Mio-Pliocene, as evidenced by the volcanic activity.

In the 4.2.2 section, we clarify the role of the crustal thickness variations due to mantle delamination and crustal thinning, to localize the deformation. In the section 4.3, we add two new short paragraphs discussing the importance of the timing of lithospheric necking and detachment. To our opinion, recently evidenced lithospheric necking of the sinking Gibraltar slab under the West Alboran Basin would have a slight to no effect on the upper plate deformation. Because the slab is imaged as non-detached, the sinking lithosphere exerts a vertical traction that is likely to be constant during the Plio-Quaternary. Mantle delamination and crustal thinning could be active processes in the Rif as discussed in Petit et al. 2015. However, this process is linked to convective lithospheric removal and may occur since upper-Miocene. A reappraisal of lithospheric delamination in the Rif around 1.8 – 1.12 Ma is dubious because lithospheric delamination in thermomechanical models show a widespread (> 100km) subsidence and fast vertical motion that we do not observe.

In the conclusion, we modify the last paragraph to leave more room to future discussions about conjoint influence of tectonic indentation and lithospheric delamination.

We believed that the present version of the manuscript would fulfil the standard for publication in Solid Earth.

With our best regards, Manfred Lafosse and co-authors.

Detailed modifications and answer to R3.

*This first sentence is confusing. I do not understand how you can determine the tectonic evolution of a region using sedimentary dynamics.*

We agree with the referee; the abstract is confusing. Recent publications about the ages of the sedimentary sequences and the sedimentary dynamics of the deep basins give us a better understanding of the evolution of the basin. It allows a better understanding of the deformation at a regional scale.

*Is this last sentence a result of your research? If that is the case you should write it in your discussion or conclusions, not here. If it is based in other authors results, please cite them conveniently.* We agree with the referee and modify sentence L36. "The Alboran Basin could be a typical example of such a complex tectonic evolution."

*No evidence of extensional or strike-slip structures active before the Tortonian are shown in this manuscript. Furthermore, the western Alboran basin, the object of this work, is not a back-arc basin. It is located in a fore-arc position, to the west of the volcanic arc (Booth-Rea et al., 2007; 2018; Gómez de la Peña et al., (2018).*

We thank the reviewer for his remark and agree that we do not show deep extensional in the present manuscript. However, Do Couto et al., (2016) and previously Comas et al., (e.g. 1999) describe a syn-rift sequence in the West Alboran Basin (WAB) for the late Aquitanian–Burdigalian to the Langhian stratigraphic units. The full geographical extent of this syn-rift sequence and particularly the extent of the Burdigalian-Langian under-compacted shales will be present in a paper currently in revision (d'Acremont et al., Tectonics, in review). In addition, Martínez-García et al., (2017) show inverted Miocene extensional structures in the South Alboran Basin, to the East of the studied area.

We thank the referee for his remarks on the forearc or back-arc position of the studied area. Onshore, the studied area is set between the Jebha and the Nekor faults, which cumulate around 290km of total left lateral displacement during the Miocene roll-back (Spakman et al., 2018). North of the Alboran Ridge, volcanism belongs to the High-K to shoshonitic series, whereas we can observe the essential volume of magmatic arc crust to the South and the East of the Alboran Ridge (Duggen et al., 2008; El Azzouzi et al., 2014). The Ras Tarf volcano consists of andesite rocks of the normal to high K – Calc alkaline series and of Miocene age (8.5 – 14 Ma). Vergés and Fernàndez, (2016) propose that the Alboran Basin is a back-arc basin. In our opinion, the studied area is in a forearc position from a petrological point of view but belongs to the left-lateral shear zone to the south of the Alboran Arc from a structural point of view. We modify the introduction accordingly by removing any mention about the position of the studied area within the arc as it still debated and has no interest in the present paper.

*Please be more precise with this age. The Tortonian represents a long time period between 11.6 and 7.2 Ma. Furthermore, the Tortonian was the period with largest tectonic subsidence (Rodríguez-Fernández et al., 1999) and arc related magmatism in the Alboran basin and also onshore (e.g. Duggen et al., 2008). Tectonic subsidence that was mostly related to extensional tectonics (Booth-Rea et al., 2004a, Giaconia et al., 2014).*

Do Couto et al., (2014, 2016) propose that a tectonic reorganization marks the tectonic inversion in the WAB and the Sorbas Basin around 8-9 Ma, during the Tortonian. According to (Giaconia et al., 2014), the extensional phase in the Betic occurred during the Serravallian to late Tortonian (13.8 to 7.5 Ma).

It is 1 Ma younger than proposed in Do Couto et al. (2014, 2016). In the South Alboran Basin (SAB), Martínez-García et al., (2017) mapped inverted Miocene structures. The authors present the Alboran Ridge as 9-11 Ma old extensional basin. It is, however, out of the scope of this paper to review the exact age of the inversion since we focus on Plio-Quaternary tectonic evolution of the studied area.

*line 105: In the WAB, a syn-rift sequence is dated from late Aquitanian–Burdigalian to the Langhian (Do Couto et al., 2016).*

*In the abstract, conclusions and discussion I miss other alternative tectonic mechanism for explaining the evolution of your studied region. Is there no other tectonic mechanism acting in the area apart from plate convergence? Many papers confer a role in the tectonics of the Western Alboran region to the underlying Alboran slab (e.g. González-Castillo et al., 2015; Mancillla et al., 2015; Heit et al. 2017, Spakman et al., 2018; Capella et al., 2019; Sun and Bezada, in press). Other alternatives to only plate convergence should be discussed in the paper.*
We agree with the referee that we can discuss our model better. Based on the papers mentioned by the referee, in the Betic and the Rif, active delamination or slab tearing might control the deformation. Based on the aforementioned papers, slab tearing propagates from the Betic to the western tip of Alboran Ridge, with 4-5 clusters of lithospheric thinning distributed from North to South below the WAB (Heit et al., 2017; Mancilla et al., 2015; Sun and Bezada, 2020). None of these papers examines the timing of the deformation, and it is not clear when this tearing process starts. We add two small paragraphs exposing our view line 415 to 428.

*The Alboran tectonic domain has been recently redefined as only the upper plate of the Betic-Rif internal zones (Booth-Rea et al., 2015; Jabaloy et al., 2018). Most of the basement of your studied region probably does not correspond to the Alboran domain, being formed by para-autochthonous low-grade metamorphic rocks of the Maghrebian Rif units or by volcanic accretion (e.g. Azdimousa et al., 2018; Gomez de la Peña et al., 2018).*

We agree with the referee and thank him for his remarks. In our manuscript, we choose to use the words Alboran tectonic domain because the Alboran Basin from north of Nekor Fault to the Internal Betic is not a rigid block as evidence by the diffuse pattern of seismicity and active faulting. Therefore, from a structural point of view it is not possible to define a rigid Alboran Tectonic block. We now clarify our choice of word; it approximatively corresponds to the *Rif-Betic*-Alboran domain of Grevemeyer et al., (2015*)*. We modify the text to clarify this point. L135-137: *At present day, GPS velocities define an Alboran tectonic domain in between African or Iberian rigid blocks (inset Fig. 1)(Grevemeyer et al., 2015; Neres et al., 2016; Palano et al., 2013, 2015). This block is limited eastward by the TASZ and by the Yusuf Fault (Fig. 1 and 2b).*

*The Carboneras fault has also been active with sinistral movement during the Plio-Quaternary (Gracia et al., 2006; Moreno et al., 2015).*

*We agree with the referee. We meant that, according to Giaconia et al. 2015, deformation might have migrated from East to West along the Trans-Alboran Shear Zone. We modify the text to avoid the confusion.*

*This discussion is not exactly a discussion. For it to be so, you need to discuss other alternative ideas to yours, which you have previously mentioned in the introduction. Here there is no mention to the tectonic effects that a deep slab could have in your studied area, for example. Please rewrite this section citing and considering other possibilities, instead of only considering plate convergence as the sole driver for deformation in the region.*

We modified the discussion according to the referee's remarks and the above arguments. However, the manuscript is already long, and we have to refrain ourselves.

[revised manuscript text omitted]

---

## Editor Decision (ED2)

[revised manuscript text omitted]

Figure entries (main tectonic events, in green):
- Central Rif: Uplift + Radial extension ↑(?)
- Onshore Nekor Fault + External Rif Units: ↓(?); Compression and uplift of the intra-mountainous Rifian basins + anticlockwise rotation ↑(?)
- Big Idrissi Volcano / NS faults: AIF early propagation ↓(?); Boudinar Basin emersion; Boudinar Basin max transgression
- SAR / NAR: Pulse 1, Pulse 2, Pulse 3
- N145° fault zone
- AIFZ Central segments: Early (re)-activation of the AIF ? ↓(?)
- Alboran Ridge: Local compressionnal reactivation
- Adra fault zone: ↓(?)
- Averoes fault: strike-Slip; NW-SE folding ↓(?)
- Yusuf fault: strike-Slip; Transtension; strike-Slip
- Abubacer Ridge: Compression

Legend: Thrust fault; Fold axis; Strike-Slip Fault; Normal Fault

**Figure 15: Synthesis of the tectonic events in the Alboran Basin, and the Rif from the literature and the present study. \*, this study; (1), Benmakhlouf et al., (2012) ; (2), Romagny et al., (2014) ; (3) Aït Brahim and Chotin, (1990), (4), Lafosse et al., (2017); (5), Azdimousa et al., (2006); (6), Galindo-Zaldivar et al., (2018); (7) Juan et al., (2016); (8) Martínez-García et al., (2013) ; (9), Martínez-García et al., (2017); (10), Gràcia et al., (2019); (11), Perea et al., (2018); (12) Giaconia et al., (2015). The main tectonic events are in green. Green arrows and question marks indicate the age uncertainties of the main tectonic events.**

850

[Figure]

(a)

⟳ vertical axis rotation

SLAB

AIF

SAR

CR

JF

NF

Early Pliocene

(b)

⟳ vertical axis rotation

SLAB

AIF

CR

JF

NF

Early Pleistocene

[Figure]

(c)

⟳ vertical axis rotation

AIF

SLAB

AIF

CR

JF

NF

Present-Day

**Figure 16: Palinspastic maps of the SAR and the Rif from 5 Ma to the present-day are using 14 ° clockwise rotation of the Alboran tectonic domain from a) to c). Dashed blue line, approximate coastline; continuous blue line, present-day coastline; Dark yellow, Miocene-Pliocene onshore basins; light yellow, Pliocene and Quaternary onshore basins; grey patch, position of the slab remaining approximatively constant below the Alboran Basin during the Plio-Quaternary. Thick grey arrows in (c) indicate the direction and relative amount of extrusion in the central Rif considering a fixed Eurasia. The shortening is accommodated through compressive structures in (a). The initiation of subsidence along the Big Al-Idrissi Volcano and the Moroccan shelf corresponds to (b), and the present–day partitioning of the deformation corresponds to (c). CR, central Rif, JF, Jebha Fault; NF, Nekor Fault; AIF, Al-Idrissi Fault zone.**

---

## Author Response (AR3)

**Cover Letter**

Dear Dr Rossetti,

In the present version of the paper entitled "Plio-Quaternary tectonic evolution of the southern margin of the Alboran Basin (Western Mediterranean)", several modifications have been made in accordance with the points you raised in your last review.

We modified the Introduction section. It has been shortened and modified to present the scientific problem clearly. We also reorganized the section to avoid the repetitions. In the result section, we modify the text to avoid scientific inferences and moved them into the discussion.

We reorganized and edited the discussion section. Table 1 has been moved to the 15$^{th}$ position of the figures. Eventually, we do not change the words 'restraining bend', which is the exact terminology for a compressive relay in between strike-slip fault segments (e.g. Mann, 2007). We modified the text to demonstrate better the transpression. Blocks and basements faults rotation are based on a comparison with analog models and the literature. The last section has been shortened according to previous comments. We modified the figures to be sure that localities are present on the maps.

Overall, we took great attention to the style and have corrected many grammatical mistakes. It results a shorter and clearer paper. We believe that this version is suitable for publication. We also would like to add a co-author (Dr Jeroen Smit) who helped us to reviewed to grammar and the organization of the text during this iteration.

Dr Manfred Lafosse and co-authors.

[revised manuscript text omitted]

| | | Ref: (1, 2) | (3, 5) | (*, 4) | (*, 4) | (*, 4, 5) | (*, 6) | (*, 7) | (8, 9) | (*, 8, 9, 10) | (10) | (10) | (10, 11) | (8, 9) | (12) |
| 0 | Pleistocene – Middle / Calabrian / Gelasian | ↓ (?) | | | | | | Local compressionnal reactivation | | | | ↓ (?) | strike - Slip | | |
| 1 | | ↑ (?) | | | | AIF early propagation ↓ (?) | | | Pulse 3 ↓ (?) | ↓ (?) | | | | | |
| 2 | Pliocene – Piacenzian | Uplift + Radial extension | ↑ (?) | | | | | Pulse 2 | | Early (re)-activation of the AIF ? | | | | Transtension | Compression |
| 3 | | | Compression and uplift of the intra-mountainous Rifian basins + anticlockwise rotation | | | ← Boudinar Basin emersion | | | | | | | | | |
| 4 | Zanclean | | | | | | | | | | | | NW -SE folding ↓ (?) | strike - Slip | |
| 5 / 5.33 M | | | | | | ← Boudinar Basin max transgression | | | Pulse 1 | | | | | | |

[revised manuscript text omitted]

-600m

-500m

-400m

-300m

-200m

-100m

Francesc Pagès Bank

Al-Idrissi fault zone

Al - Idrissi fault splay

N145° fault zone

fig. b

fig. 9

Bokkoya fault

Small Al - Idrissi volcano

Big Al - Idrissi volcano

-600m

-500m

-400m

Fig. 8

Ras Tarf

Boudinar fault

0 1 2 3 4 5 km

-3.90   -3.85   -3.80   -3.75   -3.70   -3.65   -3.60

35.65   35.60   35.55   35.50   35.45   35.40   35.35

**b)** SW     **N145° fault zone**     NE

100 mstwt

2km

1060    **Figure 11: Active structures around the roughly NNE-SSW AIF and adjacent submarine highs. The AIF bends to the North, where it follows the trends of the NAR. High values of curvatures in the Francesc Pagès Bank and the Northeast corner of the map underline the linear features at the seafloor, which corresponds to the truncated Miocene-Pliocene layers. Extreme positive values in red represent concave topography at the seafloor; extreme negative values in blue represent convex topography. a) Profile curvature map textured above the shaded bathymetry; dashed purple lines, fault tracks at the seafloor; dashed black lines, positions of the**
1065    **seismic line in (b) and in figures 8, 9, and 13. b) TOPAS profile showing active N145° normal faults. Red lines, active faults; red arrows, positions of the fault traces in (a).**

[Figure]

**Figure 12:** SPARKER seismic line showing the transgression of marine sediment (in green) over the prograding shelf of the Nekor Basin (in pink). Oldest depositional units (Pliocene) are colored in blue and the acoustic basement in grey. The Maximum Regressive Surface (MRS) is in red.

[Figure]

[Figure]

**Figure 13: Multichannel seismic lines across the left-lateral restraining bend of the Al-Idrissi fault zone showing lateral evolution of the tectonic structures in North Alboran Ridge and in the left-lateral restraining bend. a) The Al-Idrissi fault zone is a positive flower structure following the front of the Alboran Ridge. b) The Al-Idrissi fault zone is a positive flower structure distinct from the Pliocene thrusts and folds.**

1075

1080

[Figure]

**Figure 14: Active structures affecting the northern flank of Francesc Pagès and Ramon Margalef highs. a) plan curvature map overlying the shaded bathymetry; red arrows pockmarks on the seafloor; dashed black lines, seismic lines in the figures (b) and (c); dashed red lines, positions of the fault tracks. b) SPARKER seismic reflection line showing the northward continuity of N145° fault (red line). c) TOPAS seismic line showing the subsurface of the seafloor. Red arrows, positions of the faults drawn in a).**

**Figure 15: Synthesis of the tectonic events in the Alboran Basin, and the Rif from the literature and the present study. \*, this study; (1), Benmakhlouf et al., (2012) ; (2), Romagny et al., (2014) ; (3) Aït Brahim and Chotin, (1990), (4), Lafosse et al., (2017); (5), Azdimousa et al., (2006); (6), Galindo-Zaldivar et al., (2018); (7) Juan et al., (2016); (8) Martínez-García et al., (2013) ; (9), Martínez-García et**

al., (2017); (10), Gràcia et al., (2019); (11), Perea et al., (2018); (12) Giaconia et al., (2015). The main tectonic events are in green.

[Figure]

Figure 15Green arrows and question marks indicate the age uncertainties of the main tectonic events.

[Figure]

Early Pliocene

Early Pleistocene

Present-Day

1095

**Figure 16**: Palinspastic maps of the SAR and the Rif from 5 Ma to the present-day are using 14 ° clockwise rotation of the Alboran tectonic domain from a) to c). Dashed blue line, approximate coastline; continuous blue line, present-day coastline; Dark yellow, Miocene-Pliocene onshore basins; light yellow, Pliocene and Quaternary onshore basins; grey patch, position of the slab remaining approximatively constant below the Alboran Basin during the Plio-Quaternary. Thick grey arrows in (c) indicate the direction and relative amount of extrusion in the central Rif considering a fixed Eurasia. The shortening is accommodated through compressive structures in (a). The initiation of subsidence along the Big Al-Idrissi Volcano and the Moroccan shelf corresponds to (b), and the present–day partitioning of the deformation corresponds to (c). CR, central Rif, JF, Jebha Fault; NF, Nekor Fault; AIF, Al-Idrissi Fault zone.

---

## Author Response (AR4)

**Cover Letter**

Dear editor,

Please find the last version of our manuscript "Plio-Quaternary tectonic evolution of the southern margin of the Alboran Basin (Western Mediterranean)". The small modifications have been made according to your remarks. About the call of the figure 12, this figure was already called in the section 3.1. after the figure 11. We checked the reference to Crespo-Blanc et al., (2016). They propose a synthesis of the paleomagnetic data for the internal and external Rif and the Betic since 9 Ma. We also modified the discussion. We meant that the angle between the PDZ and the trend of the folds that is the stretching direction is equal to 20°. We simplified the sentence by removing the mention about the pure-shear dominated regime since we do not provide estimations of the vorticity.

We hope the paper is now suitable for the final acceptance.

Dr Manfred Lafosse and co-authors